# Discrimination between Carbapenem-Resistant and Carbapenem-Sensitive *Klebsiella pneumoniae* Strains through Computational Analysis of Surface-Enhanced Raman Spectra: a Pilot Study

Wei Liu,[a] Jia-Wei Tang,[a] Jing-Wen Lyu,[b] Jun-Jiao Wang,[a] Ya-Cheng Pan,[c] Xin-Yi Shi,[c] Qing-Hua Liu,[d] Xiao Zhang,[a] Bing Gu,[c,e] Liang Wang[a]

[a]Department of Bioinformatics, School of Medical Informatics and Engineering, Xuzhou Medical University, Xuzhou, Jiangsu, China
[b]Medical Technology School of Xuzhou Medical University, Xuzhou, Jiangsu, China
[c]School of Life Sciences, Xuzhou Medical University, Xuzhou, Jiangsu, China
[d]State Key Laboratory of Quality Research in Chinese Medicines, Macau University of Science and Technology, Taipa, China
[e]Laboratory Medicine, Guangdong Provincial People's Hospital, Guangdong Academy of Medical Sciences, Guangzhou, China

Wei Liu, Jia-Wei Tang, and Jing-Wen Lyu contributed equally to this article. Author order was determined after negotiation.

**ABSTRACT** In clinical settings, rapid and accurate diagnosis of antibiotic resistance is essential for the efficient treatment of bacterial infections. Conventional methods for antibiotic resistance testing are time consuming, while molecular methods such as PCR-based testing might not accurately reflect phenotypic resistance. Thus, fast and accurate methods for the analysis of bacterial antibiotic resistance are in high demand for clinical applications. In this pilot study, we isolated 7 carbapenem-sensitive *Klebsiella pneumoniae* (CSKP) strains and 8 carbapenem-resistant *Klebsiella pneumoniae* (CRKP) strains from clinical samples. Surface-enhanced Raman spectroscopy (SERS) as a label-free and noninvasive method was employed for discriminating CSKP strains from CRKP strains through computational analysis. Eight supervised machine learning algorithms were applied for sample analysis. According to the results, all supervised machine learning methods could successfully predict carbapenem sensitivity and resistance in *K. pneumoniae*, with a convolutional neural network (CNN) algorithm on top of all other methods. Taken together, this pilot study confirmed the application potentials of surface-enhanced Raman spectroscopy in fast and accurate discrimination of *Klebsiella pneumoniae* strains with different antibiotic resistance profiles.

**IMPORTANCE** With the low-cost, label-free, and nondestructive features, Raman spectroscopy is becoming an attractive technique with great potential to discriminate bacterial infections. In this pilot study, we analyzed surfaced-enhanced Raman spectroscopy (SERS) spectra via supervised machine learning algorithms, through which we confirmed the application potentials of the SERS technique in rapid and accurate discrimination of *Klebsiella pneumoniae* strains with different antibiotic resistance profiles.

**KEYWORDS** *Klebsiella pneumoniae*, carbapenems, surface-enhanced Raman spectroscopy, machine learning algorithm, antibiotic resistance profile

Address correspondence to Xiao Zhang, changshui@hotmail.com, Bing Gu, gb20031129@163.com, or Liang Wang, healthscience@foxmail.com.
The authors declare no conflict of interest.

Many microbial organisms are pathogenic to human beings and are able to cause infectious diseases (1). In addition, drug-resistant bacterial pathogens have been emerging due to the overuse and misuse of antibiotics (2), which leads to difficulty in bacterial control and imposes further threats upon global public health. Thus, fast and accurate detection of antibiotic-resistant bacteria is necessary for clinical treatment of bacterial infection and prevention of bacterial transmission (3). *Klebsiella pneumoniae* is

an encapsulated Gram-negative and facultative anaerobic bacterium that was first described by Edwin Klebs in 1875 and belongs to the *Enterobacteriaceae* family (4). It is also an opportunistic bacterial pathogen that causes pneumonia-derived sepsis, leading to high morbidity and mortality (5). Due to the rapid dissemination of *K. pneumoniae* in the hospital environment, it is easy for the bacterial pathogen to cause nosocomial outbreaks (6). In fact, *K. pneumoniae* is reported to be responsible for around one-third of all Gram-negative infections in the hospital (7), which makes it the second most important opportunistic enterobacterium in nosocomial and community infections just after *Escherichia coli* (8).

In recent decades, with the increasing abuse of antibiotics in clinical settings, *K. pneumoniae* shows frequent acquisition of resistance to antibiotics, which makes the nosocomial infections caused by the pathogen particularly problematic (8). For example, extended-spectrum $\beta$-lactamases (ESBLs) mediate resistance to broad-spectrum cephalosporins and aztreonam, the coding genes of which are usually found on plasmids and harbored by *K. pneumoniae* (9). The increasing use of carbapenems has led to the abundant emergence of carbapenem-resistant *K. pneumoniae* (CRKP) strains (10), while the CRKP strains are defined as being resistant to at least one of the carbapenem agents, including ertapenem, meropenem, and imipenem (11, 12). Currently, there are many mechanisms in *K. pneumoniae* for carbapenem resistance, but few antimicrobial therapy options exist for infections caused by CRKP (13). Only tigecycline, colistin, and several aminoglycosides show favorable *in vitro* activities against CRKP, which leads to the emergence of strains with colistin resistance among CRKP strains (7). Due to the difficulty of clinical treatment of CRKP infection, it is therefore important to discriminate CRKP strains from carbapenem-sensitive *Klebsiella pneumoniae* (CSKP) strains with rapidity, cost-effectiveness, and accuracy, which is essential to instruct the initial antimicrobial use and effective control of the bacterial infection (14).

Surface-enhanced Raman spectroscopy (SERS) is a nondestructive chemical analysis technique that could improve the weak signals of regular Raman spectroscopy through interactions between sample molecules and surface plasmons of nanoscale-structured metal particles (15). In particular, signal-enhancing metal nanostructures, such as silver (Ag), copper (Cu), and gold (Au), can generate a plasmon resonance electromagnetic enhancement of the stimulating light, which could greatly increase the signal level of Raman spectroscopy up to several orders of magnitude (16). However, due to the complexity of Raman spectra, traditional linear analysis is not sufficient for the data-processing procedures, while machine learning (ML) algorithms are capable of extracting important features from the sophisticated SERS spectral data sets (15, 16). Thus, SERS provides a great potential for fast and sensitive microbial detection and identification with the assistance of appropriate ML algorithms (17). At present, few studies have applied and compared machine learning methods in terms of SERS spectral analysis in order to distinguish between CSKP and CRKP strains. In this pilot study, we isolated 7 CSKP and 8 CRKP strains from clinical samples, and their SERS spectra were analyzed via eight supervised machine learning algorithms. Among these algorithms, CNN achieved high-level accuracy in predicting CSKP and CRKP strains, with area under the curve (AUC) values reaching to 99.57% and 5-fold cross validation reaching to 99.78%. Taken together, this study showed that SERS spectra combined with a deep learning algorithm could effectively distinguish CSKP strains from CRKP strains, which reinforced its potential in real-world applications, such as bacterial diagnosis and antibiotic stewardship.

## RESULTS

**Raman spectra of CRKP and CSKP strains. (i) Average Raman spectra**. Average Raman spectra with standard error bands for CRKP and CSKP strains were generated through calculating the means of signal intensities at corresponding Raman shifts via both biological and technical repeats (Fig. 1). Although spectral profiles for the two *K. pneumoniae* groups were similar, different Raman intensities and characteristic peaks

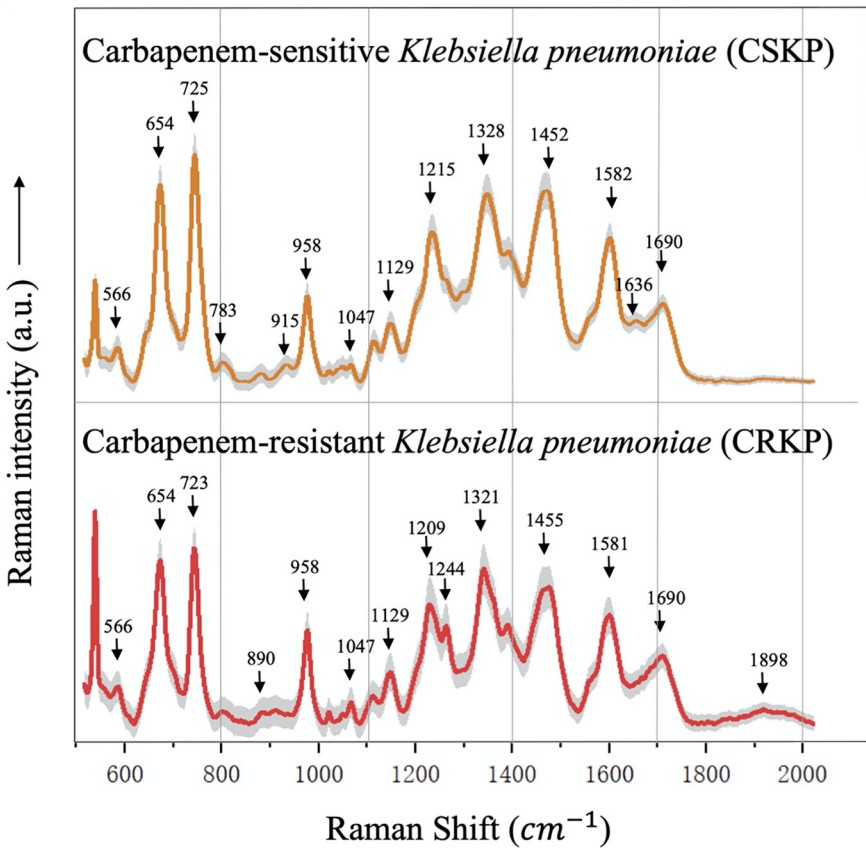

**FIG 1** Demonstration of the average SERS spectra for CSKP ($N = 280$) and CRKP ($N = 280$). Each average Raman spectrum was generated from multiple Raman spectra by calculating the mean Raman intensity at corresponding Raman shift. The characteristic peaks of each average Raman spectrum were marked with vertical black arrows. Different Raman spectra had their own combination of characteristic peaks. The shaded band of each Raman spectrum represents 20% standard error. The x axis shows the Raman shifts from 519.56 $cm^{-1}$ to 1,800.81 $cm^{-1}$, while the y axis shows the Raman spectral intensity in artificial units (a.u.).

were identified, which suggested differences in biochemistry and could be used to discriminate strains of CRKP and CSKP. In addition, the standard error bands (shaded region) quantitatively reflected the good reproducibility of Raman spectra for CRKP and CSKP strains, respectively. In order to evaluate the repeatability of Raman spectra, an average Raman spectrum with a standard error band was also generated for each *K. pneumoniae* strain, together with the distributions of characteristic peaks in a dot matrix plot (Fig. S2 in the supplemental material), according to which Raman spectra were well repeated for each strain.

Meanwhile, we also checked Raman spectra repeatability by calculating the average Raman spectrum and standard error band for each *Klebsiella pneumoniae* strain, which showed that repeatability of Raman spectra was well maintained (Fig. S2A and B).

**(ii) Characteristic peaks.** Different bacteria have their own combinations of characteristic peaks in Raman spectra due to their unique chemical compositions, which could be used to distinguish them from each other at different taxonomic levels, such as species and subspecies, etc. (18). However, raw Raman spectral data are not suitable for the identification of characteristic peaks due to the unwanted signals (noises) in the spectra (19). In order to reduce the influences of noises on the identification of characteristic peaks, we used the Savitzky-Golay (SG) smoothing filter algorithm to smooth the Raman spectral data and reduce noise interference (20). The software LabSpec 6 was then used to identify characteristic peaks with a Gaussian-Lorentzian function (21), which were marked with black arrows along the spectra (Fig. 1). In order to check

**TABLE 1** Characteristic peaks in the average Raman spectra of CSKP and CRKP strains and the corresponding chemical components

| Wavenumber (cm$^{-1}$) | Band assignment | Reference |
|---|---|---|
| 566 | Guanine/thymine/uridine | 40 |
| 654 | Guanine | 41 |
| 723/725 | Nucleic acids | 42 |
| 783 | Thymine | 43 |
| 890 | Tryptophan | 44 |
| 915 | C-C (carbon–carbon single bond) | 45 |
| 958 | C = C | 41 |
| 1,047 | P-O | 46 |
| 1,129 | CH$_2$ | 41 |
| 1,209 | Phenylalanine/tyrosine | 47 |
| 1,215 | CH | 48 |
| 1,244 | Amide III | 49 |
| 1,321 | Guanine | 50 |
| 1,328 | Adenine ring | 51 |
| 1,452/1,455 | N = N aromatic and aliphatic | 52 |
| 1,581/1,582 | Guanine/adenine | 51 |
| 1,636 | Amide I | 50 |
| 1,690 | C = O, C = C | 52 |
| 1,898 | C = O | 52 |

whether *K. pneumoniae* strains could be successfully separated solely based on the unique profiles of characteristic peaks into CSKP and CRKP groups, we also performed a principal-component analysis (PCA) for the data, which showed that all the samples could be correctly classified except for two samples, 26-272 and 20-18 (Fig. S3). Thus, characteristic peaks could not be reliably used for the classification of antibiotic resistance phenotypes.

According to previous studies, characteristic peaks in Raman spectra corresponded to different biochemical molecules (22). In addition, the more complex the chemical composition of a bacterium, the richer its Raman spectrum (23). In this study, individual chemical components represented by characteristic peaks were sourced from previous reports in the literature and summarized in Table 1. According to the results, CSKP and CRKP had the same characteristic peaks at 566 cm$^{-1}$ (guanine/thymine/uridine), 654 cm$^{-1}$ (guanine), 723/725 cm$^{-1}$ (nucleic acids), 958 cm$^{-1}$ (carbon-carbon double bond [C = C]), 1,047 cm$^{-1}$ (phosphorus-oxygen bond [P-O]), 1,129 cm$^{-1}$ (CH$_2$), 1,452/1,455 cm$^{-1}$ (nitrogen-nitrogen double bond [N = N] aromatic and aliphatic), 1,581/1,582 cm$^{-1}$ (guanine/adenine), and 1,690 cm$^{-1}$ (carbon-oxygen double bond [C = O], C = C). For details of the biological meanings and the corresponding references of all the characteristic peaks, please refer to Table 1.

**Supervised machine learning algorithms. (i) Algorithm comparison.** The purpose of supervised machine learning analysis is to construct appropriate prediction models for recognizing Raman spectra in between CSKP and CRKP strains. In this study, we compared eight supervised machine learning algorithms in terms of their capacities in predicting Raman spectra of CSKP and CRKP strains, which included convolutional neural network (CNN), gradient boosting (GB), linear discriminant analysis (LDA), *k*-nearest neighbors (KNN), random forest (RF), adaptive boosting (Adaboost), decision tree (DT), and support vector machine (SVM). The process of a supervised learning algorithm is to divide the data into independent training and test sets. Data in the training set will be labeled first, which will then be trained to obtain an optimal prediction model that will be applied to unlabeled test data and mapped into output results. The performance of each algorithm was measured by four indicators, accuracy (ACC), precision, recall, and F1 (F1 is an overall measure of a model's accuracy that combines precision and recall, a good value of which indicates low false positives and low false negatives). Cross-validation (CV) is also an efficient method for assessing effectiveness, overfitting, and stability of supervised machine learning models when sample size is

**TABLE 2** Comparison of performance measures of eight different supervised machine learning algorithms

| Algorithms | ACC[a] | Precision | Recall | F1 | 5-Fold CV[b] |
|---|---|---|---|---|---|
| CNN | 100% | 100% | 100% | 100% | 99.78% |
| GB | 99.40% | 99.40% | 99.38% | 99.40% | 94.91% |
| LDA | 99.40% | 99.40% | 99.38% | 99.40% | 81.10% |
| KNN | 98.21% | 98.21% | 98.23% | 98.21% | 94.90% |
| RF | 98.21% | 98.21% | 98.28% | 98.21% | 94.65% |
| AdaBoost | 97.62% | 97.62% | 97.57% | 97.62% | 95.17% |
| DT | 96.43% | 96.43% | 96.47% | 96.43% | 93.63% |
| SVM | 93.54% | 93.54% | 93.45% | 93.49% | 93.43% |

[a]Algorithms were ranked from high to low values of accuracy (ACC).
[b]5-Fold CV, fivefold cross validation.

small (24). Thus, in this study, we performed 5-fold cross-validation (5-fold CV) for all the supervised machine learning algorithms.

In specificity, according to the results, CNN had a prediction accuracy of 100%, and its 5-fold cross-validation reached to 99.78%, which made the algorithm best in predicting CSKP and CRKP strains compared with the other 7 supervised machine learning algorithms. As for GB and LDA, their performance measures were exactly the same (ACC = 99.4%, precision = 99.4%, recall = 99.38%, F1 = 99.4%). However, 5-fold cross-validation showed that GB (94.91%) had higher average accuracy than LDA (81.1%), which suggested that GB was more stable than LDA in terms of Raman spectral analysis. The other five algorithms also showed comparatively good performance in terms of prediction capacities, among which SVM (ACC = 93.54%) had the lowest accuracy. Taken together, CNN was found to be the best prediction model, while SVM was the worst. However, when using 5-fold CV as a measurement, LDA had the worst performance. For details of the performance measures for the eight algorithms, please refer to Table 2. Moreover, it was also noteworthy that two kernel functions of the SVM algorithm, linear function (linear) and radial basis function (rbf), were compared, according to which linear kernel function (ACC = 93.54%, precision = 93.54%, recall = 93.45%, and F1 = 93.49%) performed better than rbf kernel function (ACC = 80.36%, precision = 80.36%, recall = 80.44%, and F1 = 80.36%). Thus, SVM with linear kernel function was more appropriate for dealing with dichotomy problems than the rfb kernel function that was not included in Table 2.

**(ii) Receiver operating characteristic curves.** To measure the advantages and disadvantages of each supervised machine learning model used in this study for the prediction of CSKP and CRKP strains, receiver operating characteristic (ROC) curves were used to compare the sensitivity and specificity of the prediction results of each model (Fig. 2). The x axis represents specificity, which is also called the false-positive rate. The closer the x axis is to zero, the higher the accuracy rate. The y axis represents sensitivity, which is also known as true positive rate (sensitivity). The larger the y axis, the better the accuracy. Therefore, the closer the ROC curve is to the upper left corner, the higher the accuracy of the experiment. Meanwhile, area under curve (AUC) was also calculated for each ROC curve in order to quantitatively measure the performance of each model. The larger the value of AUC, the better the performance of the model (Fig. 2). According to the results, CNN (AUC = 0.9957) had the best performance, which was tightly followed by LDA (AUC = 0.9745) and Adaboost (AUC = 0.9767). As for other models, their AUC values were also comparatively good and were all greater than 0.95, except for SVM (AUC = 0.9414), which had the lowest AUC value.

**(iii) Confusion matrix.** A confusion matrix is a visual display used to describe the performance of a classification model on a set of test data for which the true values are known. Each column of the matrix represents the sample predicted by the model, while each row of the matrix represents the true status of the sample. In this study, we drew a set of binary classification confusion matrices for the eight supervised machine learning models (Fig. 3). Compared with other models, the results showed that the final

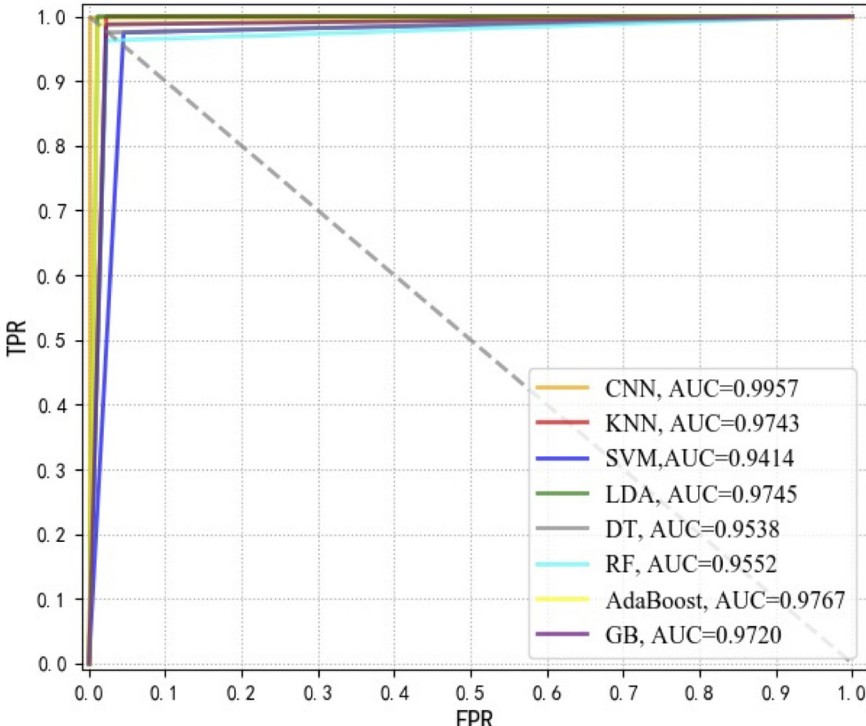

**FIG 2** ROC curves of eight supervised machine learning algorithms used in this study. Through comparison, it could be seen that the CNN model (AUC = 0.9957) had the best performance for predicting CSKP and CRKP strains in this study; TPR, true positive rate; FPR, false positive rate.

recognition accuracy of the CNN model for the two groups of Raman spectra, CSKP and CRKP, was the best and reached to 100%, which verified the feasibility of the CNN model for the prediction and classification of bacterial antibiotic-resistant and antibiotic-sensitive phenotypes based on SERS spectra.

**Effects of signal-to-noise ratio on machine learning accuracy.** During the generation of Raman spectra from bacterial samples, it was impossible to avoid the interfer-

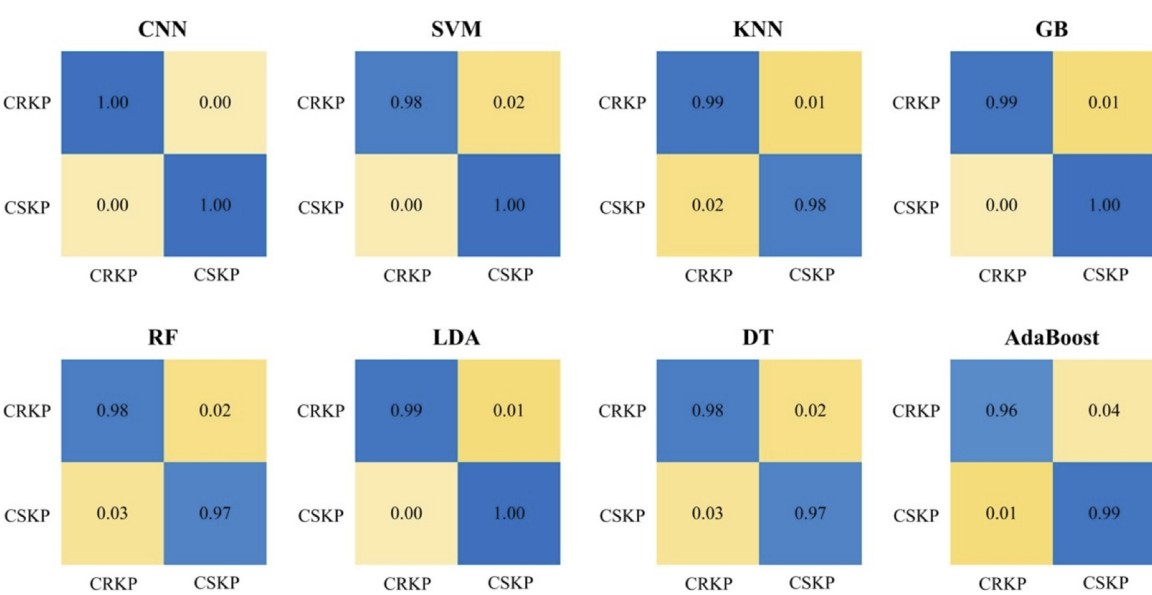

**FIG 3** Confusion matrices for eight machine learning algorithms in terms of classification of CSKP and CRKP strains. For each confusion matrix, rows correspond to phenotypes (antibiotic resistance or sensitivity) identified by standard biochemical tests (true class), while columns correspond to phenotypes predicted by supervised machine learning algorithms (predicted class). Numbers in the confusion matrix stand for the percentage of correctly classified (diagonal) or misclassified (off-diagonal) spectra, respectively.

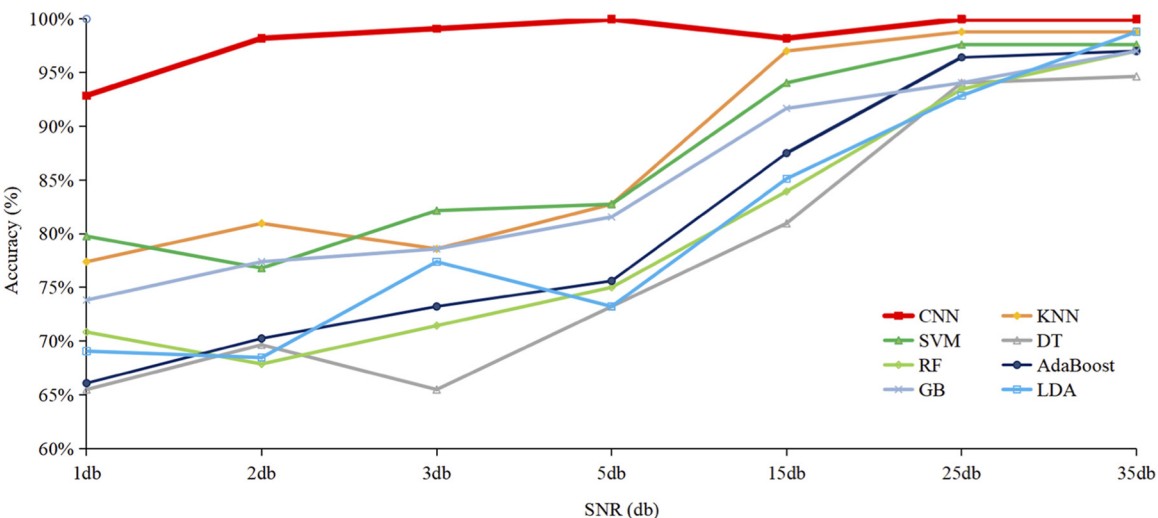

**FIG 4** Quantitative analysis of the influences of different SNR on eight supervised machine learning algorithms. As seen in the figure, with the increment of SNR added to raw Raman spectra, accuracy of classification algorithms generally increased. Among all the models, CNN showed the strongest robustness in its antinoise ability. When 1-db SNR was added to the Raman spectra, CNN could still classify Raman spectral data with an accuracy rate over 90%.

ence of many uncontrollable factors, including environmental noise, fluorescence, and radiation, etc. The existence of a variety of interference noises requires higher anti-interference ability and robustness of classification algorithm. In this study, we added artificial noises to raw Raman spectral data via Gaussian noise interference with different SNRs (1 db, 2 db, 3 db, 5 db, 15 db, 25 db, and 35 db [decibels]) and then compared the effects of SNR on the classification accuracy of eight supervised machine learning algorithms (Fig. 4). According to the results, the general trend was that the higher the SNR, the better the classification accuracy. In addition, CNN showed better stability and higher accuracy than the other seven tested algorithms. Thus, CNN had the best anti-interference ability when dealing with raw Raman spectra data.

## DISCUSSION

In recent years, multidrug-resistant (MDR), extensively drug-resistant (XDR), and pan-drug-resistant (PDR) bacterial pathogens are increasingly being reported worldwide and are not uncommon to be identified in different bacterial species, such as *Pseudomonas aeruginosa*, *Acinetobacter baumannii*, and *Klebsiella pneumoniae*, which leads to higher outbreak potentials and international spread of bacterial pathogens (25). As an opportunistic pathogen, *K. pneumoniae* is a natural inhabitant of the gut microbiota and is also commonly encountered in hospital-acquired infections (7). In addition, *K. pneumoniae* can cause many serious diseases, such as pneumonia, urinary tract infections, and bloodstream infections, while increasing numbers of strains resistant to antibiotics have been reported due to antibiotics abuse. According to the CLSI guidelines, CRKP has been defined as the first clinical *K. pneumoniae*-positive culture from inpatients with resistance to at least one of the following carbapenems: meropenem, imipenem, and ertapenem (12). Since carbapenems are the last line of defense against multidrug-resistant Gram-negative infections (26), CRKP represents a great challenge for clinical practitioners (4). Thus, rapid and accurate identification of CRKP is crucial for prescribing antibiotic therapy and relevant treatment strategies (4). Since all the CRKP strains were multidrug resistant, we checked other antibiotic-resistant phenotypes for all the CSKP and CRKP strains isolated in this study in addition to carbapenem resistance. The complete antibiotic resistance profiles are presented in Fig. S1A in the supplemental material and are detailed in Table S1 (CRKP) and Table S2 (CSKP). A PCA algorithm separated CRKP and CSKP strains well into two independent groups (Fig. S1B), while hierarchical clustering analysis (HCA) clustered CRKP and CSKP into

two different hierarchies except for one misclassification (Fig. S1A), that is, sample 16-2. The results indicated that the two groups of *K. pneumoniae* strains were intrinsically different in terms of their antibiotic resistance profiles, although capacity of statistical algorithms differed in separating the samples into corresponding groups solely based on antibiotic-resistant profiles.

Traditional methods for the detection of antibiotic resistance usually take at least 6 to 18 h for preliminary results and 48 h or longer for definitive results, which significantly delays the choice of an appropriate antimicrobial therapy (3), not even mentioning the fastidious and nonculturable bacterial pathogens. Thus, novel methods are needed for fast and reliable identification of bacterial antibiotic resistance. Although Raman spectroscopy (RS) has been considered a potential technique with label-free and noninvasive features for the analysis of bacterial pathogens, there is currently no real-world applications of RS in clinical settings for bacterial analysis due to various restrictions, such as weak Raman scattering effect and low reproducibility and repeatability, etc. (15). Currently, surface-enhanced Raman spectroscopy (SERS) has been extensively developed to overcome the weak Raman scattering effect, which uses metallic nanoparticles (gold, silver, and copper) to concentrate electromagnetic energy via surface plasmons (27), although the reproducibility and repeatability of SERS is also debatable (28). For example, Witkowska et al. systematically compared the differences between RS and SERS in terms of bacterial detection, according to which SERS spectra had much better quality than the normal Raman spectra for both *Escherichia coli* and *Bacillus subtilis* (29). In addition, low reproducibility and repeatability are caused by several uncontrollable external factors during an experiment (30), which could be partially reduced through increased number of biological and technical repeats. An averaged Raman spectrum with standard error was thus generated for the analysis of characteristic peaks (Fig. 1), which was also used as the representative spectrum of a specific bacterial strain (Fig. S2). Taken together, due to the greatly enhanced signal intensity, SERS was applied to all the *K. pneumoniae* strains for antibiotic resistance analysis in this study.

Due to the complexity of SERS spectra, traditional statistical methods are not sufficient to deal with the data analysis procedures (15). Thus, advanced computational methods such as supervised machine learning algorithms have been recruited for sample prediction. For example, Wang et al. (31) used CNN- and artificial neural network (ANN)-classified and predicted 18 *Arcobacter* species from clinical, environmental, and agri-food sources with an accuracy rate of 97.2%. In addition, Tang et al. also successfully identified a set of clinically isolated *Staphylococcus* species via the combination of surface-enhanced Raman spectral fingerprinting and machine learning algorithms, which also confirmed the potential applicability of the SERS technology in clinical diagnostics (17). In terms of the differentiation of antibiotic resistance and sensitivity in bacterial strains, a variety of studies have addressed this question. However, most of the studies used simple statistical models, such as linear discriminant analysis (LDA) and principal-component analysis (PCA) for data analysis. For example, Verma et al. used partial least squares-discriminant analysis (PLS-DA) to study Raman spectra of *Escherichia coli* strains treated with bacteriostatic and bactericidal antibiotics, which identified characteristic peaks that are altered by antibiotic concentrations (32). In addition, Cheong et al. analyzed drop-coating deposition SERS spectra via PCA and SVM to identify quinolone-resistant *K. pneumoniae* strains (33).

We applied an advanced SERS technique coupled with machine learning algorithms to clinically isolated *K. pneumoniae* strains, through which CSKP and CRKP strains were rapidly and accurately recognized. A total of eight commonly used supervised machine learning methods, including AdaBoost, CNN, DT, GB, KNN, LDA, RF, and SVM, were performed on SERS spectral data and compared in terms of their capacities in predicting CSKP and CRKP strains. Among these algorithms, CNN consistently performed best based on all the evaluation indicators (Table 2), ROC

curves (Fig. 2), and confusion matrix (Fig. 3), achieving 99.78% accuracy during 5-fold cross-validation. Previously, Ho et al. (34) used convolutional neural network (CNN) and support vector machine (SVM) methods to successfully identify methicillin-resistant *Staphylococcus aureus* (MRSA) and methicillin-sensitive *S. aureus* (MSSA) with an accuracy of 89 ± 0.1%. Thus, the pilot study showed that CNN could be used for antibiotic resistance predictions in differential bacterial species with better performance. Moreover, by comparing with other machine learning methods used in this study, CNN could handle complex regression and classification problems without assumed mathematical equations between input and output, leading to high computational efficiency and strong fault tolerance (31). So far, few studies have paid attention to how Raman spectral preprocess influenced the analysis of machine learning algorithms. Our study also compared anti-interference capacities of these algorithms in terms of artificially added noises (Fig. 4), which consistently revealed that CNN performed the best compared with other supervised machine learning algorithms. Thus, we concluded that CNN has good robustness on low signal-to-noise ratio data in the SERS spectra.

In this pilot study, we performed comparative analyses of supervised machine learning algorithms on discriminating CSKP and CRKP strains via SERS spectra. Although all the methods achieved relatively high prediction accuracies, there are still many aspects that need to be improved for the potential application of the method in clinical settings. For example, the models that we constructed were not robust and sufficient for real-world applications due to the limited number of *K. pneumoniae* strains used in this study. In addition, since both CSKP and CRKP strains were multidrug-resistant bacteria, other antibiotic resistances rather than carbapenem resistance may also be involved in the identification of CSKP and CRKP strains because of their contributions to the generation of SERS spectra. Thus, more SERS spectra from clinically isolated CSKP and CRKP strains should be used for training the machine learning models, which would greatly improve the quality and robustness of the models. In addition, antibiotic resistance profiles during *K. pneumoniae* isolation should be strictly controlled, and those strains only with differences in carbapenem sensitivity and resistance should be used for SERS spectral analysis, while the profiles of other antibiotic resistance should be the same. In this way, machine learning models could reliably predict CSKP and CRKP strains solely based on carbapenem resistance rather than other antibiotic resistances. It should be noted that, although acquisition of the signal for a single SERS spectrum took only seconds, the method used in this study still required bacterial culture and isolation, which made the overall procedure time consuming. In future studies, we will aim to use machine learning models to recognize CSKP and CRKP strains from clinical samples directly, which will greatly improve the efficiency of the rapid diagnostics of carbapenem-resistant and carbapenem-sensitive *K. pneumoniae* strains.

**Conclusion.** Surface-enhanced Raman spectroscopy has been widely studied in terms of its application potentials in the diagnosis of bacterial pathogens and detection of antibiotic resistance. In this study, we calculated the average SERS spectra for CSKP and CRKP strains, through which the profiles of their characteristic peaks were identified. We then explored supervised machine learning algorithms in terms of their capacities in predicting CSKP and CRKP strains via SERS spectra. According to the results, eight supervised machine learning methods could successfully predict carbapenem sensitivity and resistance in *K. pneumoniae*, with the CNN algorithm on top of all other methods. In addition, CNN also performed best on SERS spectra with low signal-to-noise ratios. Taken together, our study confirmed the application potentials of surface-enhanced Raman spectroscopy in fast and accurate discrimination of *K. pneumoniae* strains with different antibiotic resistance profiles.

## MATERIALS AND METHODS

**Bacterial strains.** Both CSKP (*n* = 7) and CRKP (*n* = 8) strains were directly isolated from clinical samples and cultured on Columbia blood agar plates (35℃, 18 to 24 h) at the Department of Laboratory

Medicine, Affiliated Hospital of Xuzhou Medical University. It is noteworthy that all the clinical samples were previously deidentified, and only bacterial isolates were analyzed in this study. Drug susceptibility was identified through Vitek2 Compact, an automated microbial identification (ID)/antibiotic susceptibility testing (AST) instrument (bioMérieux, La Balme-les-Grottes, France) in the Department of Laboratory Medicine, Affiliated Hospital of Xuzhou Medical University. Carbapenem resistance together with other antibiotic resistance profiles were determined according to the MIC breakpoint standards of the CLSI Subcommittee on Antimicrobial Susceptibility Testing (M100-S30) (Tables S1 and S2 in the supplemental material) (35). All bacteria were confirmed with biochemical tests and matrix-assisted laser desorption ionization–time of flight mass spectrometry (MALDI-TOF MS) for strain typing and were then stored at −80°C (Thermo Fisher, USA). In addition, principal-component analysis (PCA) and hierarchical clustering analysis (HCA) methods were applied to group these bacteria into two groups based on their antibiotic resistance profiles. Distribution patterns of antibiotic resistance in the CSKP and CRKP strains were visualized through interactive Tree of Life (iTOL) and are presented in Fig. S1 (36).

**Preparation of $AgNO_3$ solution.** $AgNO_3$ (33.72 mg; Sinopharm, Beijing, China) was weighed and gently mixed with 200 mL of deionized water ($ddH_2O$) in a clean sterile triangular flask, which was then heated on a magnetic stirrer (ZNCL-BS230, Shi-Ji-Hua-Ke Pty. Ltd., Beijing, China) until boiling; then, 8 mL of 1% (wt) sodium citrate was added into the mixture and stirred with a speed of 650 r/min. Heating was stopped, and stirring was continued until the mixture cooled to room temperature (RT). The final volume was set to 200 mL via addition of $ddH_2O$. One milliliter of the above-made solution was transferred to a 1.5-mL Eppendorf tube and centrifuged at 7,000 r/min for 7 min (centrifuge 5430 R, Eppendorf, USA); the supernatant was discarded after centrifugation, and the pellet was resuspended with 100 $\mu$L of $ddH_2O$ to get a uniform milky gray solution. The solution is the negatively charged silver nanoparticle (AgNP) substrate. The solution was stored in the dark at RT for later use.

**Surface-enhanced Raman spectroscopy.** After each *K. pneumoniae* strain was cultured on the agar plate overnight, a single colony was picked and inoculated into 15 $\mu$L of phosphate-buffered saline (PBS) with vigorous vortexing. Fifteen microliters of negatively charged AgNPs was then mixed with the PBS solution, which was then dropped onto a silicon wafer to air dry. The dried spot was then measured via a commercial i-Raman Plus Raman spectrometer BWS456-785H (B&W Tek, USA). The measurement settings were set as follows: helium-neon (HeNe) laser power, 20 mW; wavelength, 785 nm; detector type, high quantum efficiency charge-coupled device (CCD) array; Raman shift range, 175 to 2,700 $cm^{-1}$; spectral acquisition, 5 s; resolution, <3.5 $cm^{-1}$ at 912 nm. The software BWSpec (version 4.10) was used to generate Raman spectral data. Each spectrum consisted of 657 points measured in the range of 519.56 $cm^{-1}$ to 1,800.81 $cm^{-1}$. A total of 15 *Klebsiella pneumoniae* strains were included in this experiment, which includes 7 strains of CSKP (N = 280) and 8 strains of CRSP (N = 280). Thus, a total of 560 surface-enhanced Raman spectra were collected, which was denoted by the letter N within the parentheses for each group of *K. pneumoniae* strains. For details, please refer to Table S3.

**Raman spectra data analysis.** Raman spectral analysis requires pretreatment of raw data in order to improve signal-to-noise ratio (SNR) and normalize spectral distributions, which includes curve smoothing and denoising, baseline correction, and spectral normalization. In particular, averaged Raman spectra were generated for CSKP and CRKP strains by calculating the averaged value of intensity with artificial units (a.u.) at each Raman shift in the range of from 519.56 $cm^{-1}$ to 1,800.18 $cm^{-1}$, respectively. LabSpec 6 (HORIBA Scientific, Japan) was then used for processing and smoothing the averaged Raman spectra. Characteristic peaks were calculated by following the following steps: (i) a "smoothing" function was used to smoothen averaged Raman spectra (degree of 4, size of 5, and height of 50); (ii) for baseline correction, the parameters type = polynom, degree = 6, and attach = NO were set, and "Auto" was selected to start searching for the characteristic peaks; (iii) LabSpec 6 was used to normalize the spectral data automatically in order to better compare the two curves of CSKP and CRKP; and (iv) the GaussLoren () function was used to search characteristic peaks with a level of 0% and size of 32, while other parameters were kept at default. All characteristic peaks were marked with black arrows. We then used the software Origin to generate error bands for the two averaged Raman spectra, which were based on the 20% standard deviation of Raman effect intensity corresponding to each Raman shift and could reflect the reproducibility of the experiment. An averaged Raman spectrum with a 20% standard error band together with characteristic peaks were also generated for each *K. pneumoniae* strain, which showed the repeatability of Raman spectroscopy for a single *K. pneumoniae* strain (Fig. S2). In addition, a PCA was performed based on the distribution of characteristic peaks of each bacterial strain in order to separate all the *K. pneumoniae* strains into different groups (Fig. S3).

**Machine learning algorithms. (i) Data preprocessing.** Before machine learning analysis, the software Unscrambler X was used to perform baseline correction, smoothing, denoising, and normalization of the original Raman spectra (37). In this study, we used the Savitzky-Golay (S-G) smoothing filter for denoised smoothing of Raman spectra, in which the polynomial fitting order was set to 2. For baseline correction, the multiple scattering correction (MSC) method was used, which could effectively eliminate the scattering effect of the spectral data and enhance the spectral absorption information related to the molecular compositions (38). Normalization is a method for simplifying calculation, and a variety of methods have been developed to normalize the Raman spectrum (24). In this study, we normalized SERS spectra by column (Raman intensity values at a particular Raman shift). That is, the highest intensity value (peak value) in each column was selected as the maximum constant so that all the other measured spectral intensities in the same column were divided to the highest intensity value to realize the normalization of the spectral data (39).

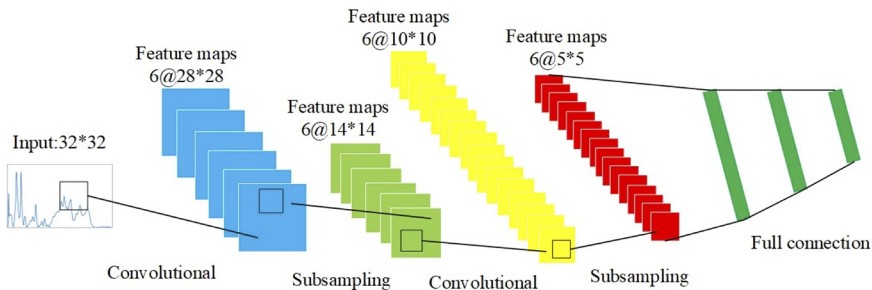

**FIG 5** Schematic illustration of LeNet-5 neural network architecture. LeNet-5 neural network classified different types of data through convolution and pooling steps and then via the full connection layer. The SoftMax activation function was finally used for the output layer.

**(ii) Supervised machine learning.** In this study, eight supervised machine learning methods, that is, adaptive boosting (AdaBoost), convolutional neural network (CNN), decision tree (DT), gradient boosting (GB), *k*-nearest neighbors (KNN), linear discriminant analysis (LDA), random forest (RF), and support vector machine (SVM), were compared for their capacities in classifying and predicting Raman spectral data by using the Python machine learning package "sklearn" (https://scikit-learn.org). Among the eight supervised machine learning algorithms, CNN is an artificial neural network, and the weights in CNN are trained through the backpropagation algorithm to achieve deep learning analysis. In this study, we used LeNet-5, a classical and efficient neural network model, for Raman spectral analysis. The schematic illustration of the network structure is shown in Fig. 5.

**(iii) Evaluation of supervised machine learning algorithms.** In order to compare the classification ability of different machine learning algorithms on Raman spectrum data, we need an evaluation standard to measure the generalization ability of the model. In the identification of spectral signals, the most commonly used performance measurements are accuracy (acc) and error rate (error), which has the following relationship: acc = 1 – error. In the evaluation of machine learning models, precision (P) and recall (R) are a pair of mutually restrictive performance metrics. Normally, the precision rate and recall rate are measurements of predictive performance. When P is high, R is low and vice versa. Therefore, when evaluating the model, in order to more intuitively reflect the performance of the model, F1 is used as a metric, which is based on the harmonic average of precision and recall (24). Because, in this study, sample size is small, when the data are divided, overfitting may occur due to unbalanced data division. For the CNN model, overfitting is more likely to occur (24). For the optimal model, we used cross-validation to divide the data set, average the results of multiple evaluations, and eliminate the adverse effects caused by the unbalanced data division, which is easier to reflect on small data sets (Fig. 6).

**Construction of confusion matrix.** A confusion matrix aims to summarize the performance of a machine learning algorithm. During the construction of a confusion matrix for each supervised machine learning algorithm, the CNN model was built on Keras architecture while the other seven supervised algorithms directly call the classifiers in the scikit-learn package, which included KNeighborsClassifier(), SVC(), DecisionTreeClassifier(), RandomForestClassifier(), AdaBoostClassifier(), GradientBoostingClassifier(), and LinearDiscriminantAnalysis(), respectively.

**Effects of SNR on machine learning accuracy.** In order to improve the performance of the model, data enhancement is usually adopted to expand the sample data and enhance the diversity of data. In this experiment, seven random SNR Gaussian white noises with different intensity were added to the Raman spectral data. Then, eight supervised machine learning models trained in this study were used

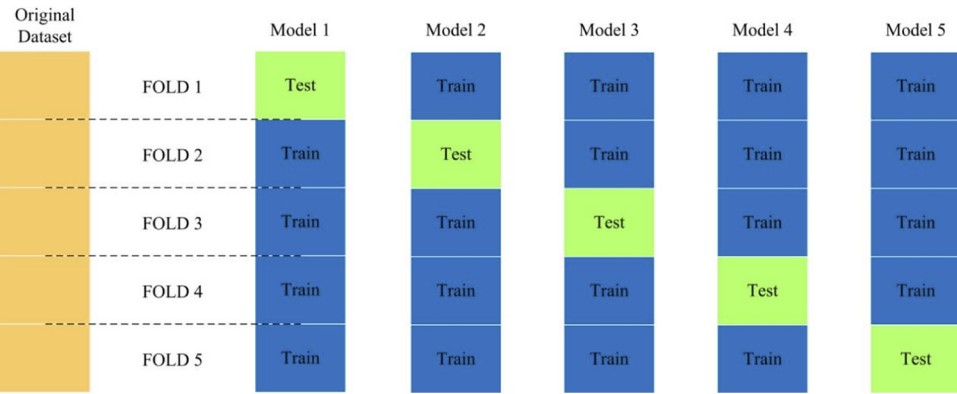

**FIG 6** Schematic illustration of 5-fold cross-validation for supervised machine learning algorithms. The 5-fold cross-validation divided the data into 5 parts and selected 1 part as the test set and the other 4 parts as the training set at each time. The above steps were repeated five times, and different parts were selected as the test set at each time, which would provide average model accuracy, indicating how stable each model was.

for testing the effects of different SNRs on the classification accuracy by following the same procedures as described above.

## SUPPLEMENTAL MATERIAL

Supplemental material is available online only.

**SUPPLEMENTAL FILE 1**, PDF file, 5.3 MB.

## ACKNOWLEDGMENTS

We thank the anonymous reviewers for their thoughtful comments that greatly improved the quality of the manuscript.

L.W. gratefully acknowledges the financial support of National Natural Science Foundation of China (31900022), Natural Science Foundation of Jiangsu Province (BK20180997), Young Science and Technology Innovation Team of Xuzhou Medical University (TD202001), and Jiangsu Qing-Lan Project (2020). B.G. thanks the financial support of the National Natural Science Foundation of China (81871734, 82072380), Key R & D Program of Jiangsu Province (BE2020646), and Research Foundation for Advanced Talents of Guandong Provincial People's Hospital (KJ012021097).

L.W., X.Z., B.G., and W.L. conceived and designed the experiments. L.W., X.Z., B.G., and W.L. contributed to the project administration. L.W., J.W.T., J.W.L., J.J.W., X.Y.S., and Y.C.P. carried out the computational and experimental work. L.W., J.W.T., and Q.H.L. wrote and revised the manuscript. L.W., X.Z., and B.G. provided the platform, resources, and student supervision. All authors read and approved the final manuscript.

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
