## [Reviewer comments · Microbiology Spectrum]

Microbiology Spectrum

Discrimination between carbapenem-resistant and carbapenem-sensitive *Klebsiella pneumoniae* strains through computational analysis of surface enhanced Raman spectra: a pilot study

Wei Liu, Jia-Wei Tang, Jing-Wen Lyu, Jun-Jiao Wang, Ya-Cheng Pan, Xin-Yi Shi, Qing-Hua Liu, Xiao Zhang, Bing Gu, and Liang Wang

Corresponding Author(s): Liang Wang, Xuzhou Medical University

Review Timeline:

Submission Date:	November 26, 2021
Editorial Decision:	December 8, 2021
Revision Received:	December 18, 2021
Editorial Decision:	December 27, 2021
Revision Received:	December 30, 2021
Accepted:	January 5, 2022

Editor: Karen Carroll

Reviewer(s): Disclosure of reviewer identity is with reference to reviewer comments included in decision letter(s). The following individuals involved in review of your submission have agreed to reveal their identity: Katarína Rebrošová (Reviewer #1)

Transaction Report:

DOI: <https://doi.org/10.1128/spectrum.02409-21>

December 8, 2021

Dr. Liang Wang
Xuzhou Medical University
School of Medical Informatics
No.209, Tongshan Road
Yunlong District
Xuzhou, Jiangsu 221000
China

Re: Spectrum02409-21 (Discrimination between carbapenem-resistant and carbapenem-sensitive *Klebsiella pneumoniae* strains through computational analysis of surface enhanced Raman spectra: a pilot study)

Dear Dr. Liang Wang:

Thank you for submitting your manuscript to Microbiology Spectrum. Your manuscript has been reviewed by two experts in the field. I concur with their assessment that the paper is of interest to readers and when revised will contribute to the literature. I also agree that the paper in its current form is too long. When submitting your revised manuscript, please reduce the length by at least 25%.

Link Not Available

Sincerely,

Karen Carroll

Journals Department
Reviewer comments:

Reviewer #1 (Comments for the Author):

The pilot study shows potential of surface enhanced Raman spectroscopy for differentiation between *Klebsiella pneumoniae* strains susceptible and resistant to carbapenems. It compares 3 unsupervised and 8 supervised machine learning methods. Supervised machine learning method allowed the authors to differ between CSKP and CRKP successfully. The presented method might extend actual applications of Raman spectroscopy in microbiology/medicine, however, further studies are needed. The article is unnecessarily long, which might discourage readers from reading it whole. There are several repetitions of the texts/ideas and too many details on (statistical) methods, which could be moved to supplementary information. The authors suggest that the method could contribute to rapid clinical diagnostic of antimicrobial resistance, but the importance of the study

in this area is not sufficiently highlighted, the article focuses more on statistical methods.
I'm adding some more comments below:

Graphical abstract:

Although graphical abstract looks catchy at the first sight, it has many components, which makes it difficult to read. I'm afraid that some texts and numbers will be too small. Plus, the meaning of the graphical visualizations might be unclear (e.g. two bottom charts).

Highlights:

The highlights draw attention to the most important findings. I would suggest not to use so many abbreviations, it might be more comfortable for a reader to write the names in full.

Methods:

- Why did authors acquire different numbers of spectra/strain? The differences are big (e.g. 20-30 vs 60 spectra).
- Why did authors use different procedures of spectral processing for machine learning methods and visualization of averaged Raman spectra?
- Why is SERS used for the measurement? If you need to culture microorganisms and you have them on agar plates, why you needed to put a colony into the solution and combine bacteria with AgNP substrate? Is there any advantage or added value? Have the authors performed also direct measurements from colonies on agar plates? It might be interesting to compare them with SERS measurements.
- Why was normalization to the highest peak chosen? Which peak was the highest or does it differ for individual spectra?
- Why the authors used unsupervised machine learning methods? They are commonly (as the authors declare as well) used for datasets without prior knowledge of all samples. Did authors try it that way, e.g. dividing the dataset to "known" and "unknown" (purposely "unidentified")?
- Suppl. Fig 1: color coding is not 100% clear - authors use 3 levels of resistance, but four colors... Also, the color coding of rectangles dividing groups is a bit confusing - I suppose, most of the readers would expect green colored strains to be sensitive and red to be resistant, as in squares further in the chart...

Results:

- Fig 3: I would recommend the authors to add labels to the arrows so the differences are clearly markable.
- Authors state: "As for CSKP strains, unique characteristic peaks were identified at 783cm⁻¹(Thymine), 915660 cm⁻¹ (C-C), 1328cm⁻¹ (Adenine ring) and 1636cm⁻¹ (Amide I), while unique characteristic peaks for CRKP strains were found at 890 cm⁻¹(Tryptophane), 1209 cm⁻¹(Phenylalanine/Tyrosine), 1244 cm⁻¹(Amide III), 1321 cm⁻¹(Guanine) and 1898 cm⁻¹(C=O)." - since the peak are marked as "unique characteristic peaks for CSKP/CRKP", how do authors explain their absence in CRKP/CSKP? Is there a possibility they are not correctly assigned to biomolecules? Or do the CRKP/CSKP strains have at least corresponding low intensity peaks? I believe the cells would need those components to survive...
- What is meant by accuracy and precision in the study? Do they apply to individual spectra or strains?
- How were the ellipses in PCA analysis depicted? Do they have any statistical relevance or were they drawn manually to include all points? (it also applies to suppl. figures)
- Were the incorrectly identified spectra acquired from one strain or more strains in each classification method?

Discussion

- Do authors believe that their method could distinguish between carbapenem-resistant and multidrug-resistant KP strains? Do authors think that the antibiotic profile as a whole reflects in Raman spectra or are the changes caused solely by carbapenem resistance?
- How would author explain the big difference between accuracies of unsupervised and supervised machine learning methods?
- Discussion should include comparison of the results with other Raman-based studies of antimicrobial resistance in microbes.
- What do the authors plan for the future - do authors plan to perform more robust analysis of CRKP and CSKP strains?
- What are the weak points of this study and how they could be improved?
- How could the results contribute rapid diagnostic testing? At the moment, method seems to be time-consuming (necessary cultivation, further sample preparation for SERS, drying time), requiring trained personal and specific consumables. Could authors comment on this? What is the approximate time necessary for the identification?

Additional comments:

- Abstract: "Conventional methods for 117 antibiotic resistance testing are sometimes time consuming while molecular methods..." - I wouldn't say sometimes since it is a bit confusing. Actually, they are time-consuming (in comparison to suggested method).
- The article is extremely long, which could prevent readers from reading it whole - for example, the introduction is 4.5 pages long and contains a lot of information that are not necessarily relevant for the study or might fit better in the discussion. Also, several detailed descriptions from methods could be moved to the supplement. Plus, the texts are repeated in many sections...
- Suppl. Fig. 1: Sentence "It was noteworthy that sample No. 16-2 was actually CRKP, which was mistakenly classified into the group of CSKP singly based on its antibiotics resistant phenotype." is not clear to me - since it had ATB resistant phenotype, why it was classified as CSKP based on it?
- Suppl. Table 3: why are demographic data (gender, age) included? Are they relevant to the study? Were the samples anonymized?

Reviewer #2 (Comments for the Author):

The article aims to demonstrate efficiency of different machine learning algorithms to discriminate among various strains of carbapenem-sensitive *Klebsiella pneumoniae* (CSKP) and carbapenem-resistant *Klebsiella pneumoniae* (CRKP) isolated from clinical samples. The data analysis of SERS spectra is well described and machine learning results are promising. In particular, according to the study, the authors identified unique profiles of characteristic peaks in average SERS spectra for CSKP and CRKP strains, respectively. In addition, they also concluded that convolutional neural network (CNN) could discriminate between CSKP and CRKP strains with high accuracy. There are also some other interesting findings reported during the study. Although the reviewer finds the study very interesting and promising, there are some issues that need to be addressed before the manuscript could be considered for publication in *Microbiology Spectrum*.

1. It would be better if the authors could add specific Raman shift for each black arrow in Figure 3 so it is much clearer to compare the differences of characteristic peaks between CSKP and CRKP strains. In addition, there should be figure legend showing which curve corresponded to which Kp strain, CSKP or CRKP.
2. Could the authors explain why they choose to use the three clustering algorithms for CSKP and CRKP analysis?
3. Could the authors provide explanations for the capacity of CNN algorithm in terms of its better stability and accuracy for low SNR SERS spectra?
4. Please make sure that all the references in the reference list follow the same format. For example, please use full name or abbreviation for all the journal names.
5. Have the authors tried other types of Raman spectrometer for the experiment, such as handheld Raman spectrometer, which will be more convenient in detecting bacterial pathogens as a POCT device and have more applications in real-world situations?
6. The methods and materials section is a bit long and the authors may want to shorten this section, especially for the routine procedures of Raman spectral preprocessing.

Staff Comments:

Preparing Revision Guidelines

For complete guidelines on revision requirements, please see the journal Submission and Review Process requirements at <https://journals.asm.org/journal/Spectrum/submission-review-process>. **Submissions of a paper that does not conform to *Microbiology Spectrum* guidelines will delay acceptance of your manuscript.** "

Please return the manuscript within 60 days; if you cannot complete the modification within this time period, please contact me. If you do not wish to modify the manuscript and prefer to submit it to another journal, please notify me of your decision immediately so that the manuscript may be formally withdrawn from consideration by *Microbiology Spectrum*.

Thank you for submitting your paper to *Microbiology Spectrum*.

Review

Discrimination between carbapenem-resistant and carbapenem-sensitive *Klebsiella pneumoniae* strains through computational analysis of surface enhanced Raman spectra: a pilot study

The pilot study shows potential of surface enhanced Raman spectroscopy for differentiation between *Klebsiella pneumoniae* strains susceptible and resistant to carbapenems. It compares 3 unsupervised and 8 supervised machine learning methods. Supervised machine learning method allowed the authors to differ between CSKP and CRKP successfully. The presented method might extend actual applications of Raman spectroscopy in microbiology/medicine, however, further studies are needed.

The article is unnecessarily long, which might discourage readers from reading it whole. There are several repetitions of the texts/ideas and too many details on (statistical) methods, which could be moved to supplementary information. The authors suggest that the method could contribute to rapid clinical diagnostic of antimicrobial resistance, but the importance of the study in this area is not sufficiently highlighted, the article focuses more on statistical methods.

I'm adding some more comments below:

Graphical abstract:

Although graphical abstract looks catchy at the first sight, it has many components, which makes it difficult to read. I'm afraid that some texts and numbers will be too small. Plus, the meaning of the graphical visualizations might be unclear (e.g. two bottom charts).

Highlights:

The highlights draw attention to the most important findings. I would suggest not to use so many abbreviations, it might be more comfortable for a reader to write the names in full.

Methods:

- Why did authors acquire different numbers of spectra/strain? The differences are big (e.g. 20-30 vs 60 spectra).
- Why did authors use different procedures of spectral processing for machine learning methods and visualization of averaged Raman spectra?
- Why is SERS used for the measurement? If you need to culture microorganisms and you have them on agar plates, why you needed to put a colony into the solution and combine bacteria with AgNP substrate? Is there any advantage or added value? Have the authors performed also direct measurements from colonies on agar plates? It might be interesting to compare them with SERS measurements.
- Why was normalization to the highest peak chosen? Which peak was the highest or does it differ for individual spectra?
- Why the authors used unsupervised machine learning methods? They are commonly (as the authors declare as well) used for datasets without prior knowledge of all samples. Did authors try it that way, e.g. dividing the dataset to "known" and "unknown" (purposely "unidentified")?
- Suppl. Fig1: color coding is not 100% clear – authors use 3 levels of resistance, but four colors... Also, the color coding of rectangles dividing groups is a bit confusing – I suppose, most of the readers would expect green colored strains to be sensitive and red to be resistant, as in squares further in the chart...

Results:

- Fig 3: I would recommend the authors to add labels to the arrows so the differences are clearly markable.
- Authors state: “As for CSKP strains, unique characteristic peaks were identified at 783cm⁻¹(Thymine), 915660 cm⁻¹ (C-C), 1328cm⁻¹ (Adenine ring) and 1636cm⁻¹ (Amide I), while unique characteristic peaks for CRKP strains were found at 890 cm⁻¹(Tryptophane), 1209 cm⁻¹(Phenylalanine/Tyrosine), 1244 cm⁻¹(Amide III), 1321 cm⁻¹(Guanine) and 1898 cm⁻¹(C=O).“ – since the peak are marked as “unique characteristic peaks for CSKP/CRKP“, how do authors explain their absence in CRKP/CSKP? Is there a possibility they are not correctly assigned to biomolecules? Or do the CRKP/CSKP strains have at least corresponding low intensity peaks? I believe the cells would need those components to survive...
- What is meant by accuracy and precision in the study? Do they apply to individual spectra or strains?
- How were the ellipses in PCA analysis depicted? Do they have any statistical relevance or were they drawn manually to include all points? (it also applies to suppl. figures)
- Were the incorrectly identified spectra acquired from one strain or more strains in each classification method?

Discussion

- Do authors believe that their method could distinguish between carbapenem-resistant and multidrug-resistant KP strains? Do authors think that the antibiotic profile as a whole reflects in Raman spectra or are the changes caused solely by carbapenem resistance?
- How would author explain the big difference between accuracies of unsupervised and supervised machine learning methods?
- Discussion should include comparison of the results with other Raman-based studies of antimicrobial resistance in microbes.
- What do the authors plan for the future - do authors plan to perform more robust analysis of CRKP and CSKP strains?
- What are the weak points of this study and how they could be improved?
- How could the results contribute rapid diagnostic testing? At the moment, method seems to be time-consuming (necessary cultivation, further sample preparation for SERS, drying time), requiring trained personal and specific consumables. Could authors comment on this? What is the approximate time necessary for the identification?

Additional comments:

-Abstract: “Conventional methods for 117 antibiotic resistance testing are sometimes time consuming while molecular methods...” – I wouldn’t say sometimes since it is a bit confusing. Actually, they are time-consuming (in comparison to suggested method).

-The article is extremely long, which could prevent readers from reading it whole – for example, the introduction is 4.5 pages long and contains a lot of information that are not necessarily relevant for the study or might fit better in the discussion. Also, several detailed descriptions from methods could be moved to the supplement. Plus, the texts are repeated in many sections...

-Suppl. Fig. 1: Sentence “It was noteworthy that sample No. 16-2 was actually CRKP, which was mistakenly classified into the group of CSKP singly based on its antibiotics resistant phenotype.” is not clear to me – since it had ATB resistant phenotype, why it was classified as CSKP based on it?

-Suppl. Table 3: why are demographic data (gender, age) included? Are they relevant to the study? Were the samples anonymized?

Point-by-Point Responses to Editorial and Reviewers' Comments

Editorial Comments

Dear Dr. Liang Wang:

Thank you for submitting your manuscript to Microbiology Spectrum. Your manuscript has been reviewed by two experts in the field. I concur with their assessment that the paper is of interest to readers and when revised will contribute to the literature. I also agree that the paper in its current form is too long. When submitting your revised manuscript, please reduce the length by at least 25%.

Many thanks for the editorial comments. We acknowledge that the manuscript is too long and have therefore shortened the length of the manuscript systematically by more than 25% of its current form. For details, please refer to the re-submitted manuscript in the submission system.

When submitting the revised version of your paper, please provide (1) point-by-point responses to the issues raised by the reviewers as file type "Response to Reviewers," not in your cover letter, and (2) a PDF file that indicates the changes from the original submission (by highlighting or underlining the changes) as file type "Marked Up Manuscript - For Review Only". Please use this link to submit your revised manuscript - we strongly recommend that you submit your paper within the next 60 days or reach out to me. Detailed instructions on submitting your revised paper are below. <https://spectrum.msubmit.net/cgi-bin/main.plex?el=A7QF4BucF3A4DksE1F2A9ftdvEsT3Y2mGBo3MB05TDT1wZ>

1. *Point-by-point responses* document was submitted as file type "Response to Reviewers"
2. PDF file indicating changes from original submission was submitted as file type "Marked Up Manuscript - For Review Only"

Thank you for the privilege of reviewing your work. Below you will find instructions from the Microbiology Spectrum editorial office and comments generated during the review. The ASM Journals program strives for constant improvement in our submission and publication process. Please tell us how we can improve your experience by taking this quick Author Survey.

The Author Survey has been completed as suggested.

Sincerely,
Karen Carroll
Editor, Microbiology Spectrum
Journals Department
Many thanks for the editorial comments that greatly improve the structure and quality of the manuscript.

Reviewers' comments

Reviewer #1 (Comments for the Author):

The pilot study shows potential of surface enhanced Raman spectroscopy for differentiation between *Klebsiella pneumoniae* strains susceptible and resistant to carbapenems. It compares 3 unsupervised and 8 supervised machine learning methods. Supervised machine learning method allowed the authors to differ between CSKP and CRKP successfully. The presented method might extend actual applications of Raman spectroscopy in microbiology/medicine, however, further studies are needed.

Many thanks for the reviewer's comments that we completely acknowledged. We have decided to remove the unsupervised machine learning part in the manuscript since it can only tell the clustering effects without facilitating the recognition and discrimination of antibiotic resistances in CSKP and CRKP strains.

The article is unnecessarily long, which might discourage readers from reading it whole. There are several repetitions of the texts/ideas and too many details on (statistical) methods, which could be moved to supplementary information.

In fact, the same issue was also pointed out by the editorial office and the other reviewer. We took the reviewer's advice, went through the manuscript, re-organized the overall structure and deleted repetitions of the texts and ideas. Therefore, the manuscript was shortened substantially. In addition, since the statistical analysis of the methods is not necessarily needed for the study and the mathematical explanation of these methods did not fit into the scope of the journal, we removed them from the manuscript.

The authors suggest that the method could contribute to rapid clinical diagnostic of antimicrobial resistance, but the importance of the study in this area is not sufficiently highlighted, the article focuses more on statistical methods.

We agree. Thus, in the revised manuscript, we focused more on the rapid clinical diagnostics of antimicrobial resistance and removed most of statistical method explanations.

I'm adding some more comments below:

Graphical abstract:

Although graphical abstract looks catchy at the first sight, it has many components, which makes it difficult to read. I'm afraid that some texts and numbers will be too small. Plus, the meaning of the graphical visualizations might be unclear (e.g. two bottom charts).

Since the graphic abstract is not required by the journal according to the authors' guidelines, in order to avoid any confusion, we removed the graphical abstract from the manuscript.

Highlights:

The highlights draw attention to the most important findings. I would suggest not to use so many abbreviations, it might be more comfortable for a reader to write the names in full.

We have changed the highlights and used full names for the abbreviations as detailed below.

Highlights

1. Carbapenem-sensitive *K. pneumoniae* (CSKP) and carbapenem-resistant *K. pneumoniae* (CRKP) strains had unique profiles of characteristic peaks in corresponding average surface-enhanced Raman spectroscopy (SERS) spectra.
2. Convolutional neural network (CNN) can discriminate between CSKP and CRKP strains

with highest prediction accuracy.

3. CNN performed best on SERS spectra with low signal-to-noise ratio than other machine learning algorithms.

Methods:

- Why did authors acquire different numbers of spectra/strain? The differences are big (e.g. 20-30 vs 60 spectra).

For some of the strains, when we used SERS technique to generate the spectra, we found that the reproducible quality of these spectra was not very good during the experiment. Thus, we tried to do more measurements for the strain to increase the repeatability of the spectra. However, considering that the sensitivity of the SERS technique, it is uncertain whether it is caused by the experimental conditions or the difference of the strains themselves. In order to make sure that our results were reliable, we used all the measured data for the analysis. That is why there were different numbers of spectra for different strains. Later during the data analysis stage, we pooled all the CSKP strains together and also CRKP strains together. We focused on the analysis of the two groups rather than specific strain in each group. We also tried to remove spectra from the strains and used the same number of spectra for each strain to do the analysis, the results of which showed no obvious effects on the overall results for machine learning analysis.

- Why did authors use different procedures of spectral processing for machine learning methods and visualization of averaged Raman spectra?

Thank you for the reviewer's question. The specific reasons for us to use different methods are present below. LabSpec is a data acquisition and analysis software designed by Horiba Scientific for the Raman spectrometers developed by the company. It has an intuitive graphic user interface (GUI) and a concise layout, which is convenient for users to conduct microscopic observations of the sample, parameter settings and data analysis for fitting characteristic peaks. However, it cannot perform batch processing on spectral data. That is why we only used the software for averaged SERS spectral analysis and identification of characteristic peaks. In contrast, the Unscrambler software does not have the function of searching for characteristic peaks, but it can preprocess the spectral data in batches. There are a variety of preprocessing algorithms for the users to choose, which is more suitable for dealing with large number of SERS spectra. Since machine learning algorithms required preprocessing of all the SERS spectra, we chose to use Unscrambler for the data preprocess.

- Why is SERS used for the measurement? If you need to culture microorganisms and you have them on agar plates, why you needed to put a colony into the solution and combine bacteria with AgNP substrate? Is there any advantage or added value? Have the authors performed also direct measurements from colonies on agar plates? It might be interesting to compare them with SERS measurements.

Many thanks for the reviewer's comments. The most important reason for us to use SERS for the measurement is because the inherent signal of Raman effect (elastic scattering) from normal Raman spectroscopy is very weak. By mixing silver nanoparticle (AgNPs) that were produced through the reactions of silver nitrate and sodium citrate with bacterial solutions, the low signal (sensitivity) problem of ordinary Raman spectroscopy could be overcome. In particular, SERS

is an enhanced RS through sample molecules interacting with surface plasmons of nanoscale structured metal surfaces, which often uses spherical nanoparticles made of silver or gold with diameters ranging from 20 to 100 nm. The amplification of the SERS signals occurs via the formation of charge transfer complex between the analyte and the SERS substrate (silver nanoparticles, copper nanoparticles, or gold nanoparticles), which produces large amplifications of the laser field (doi: 10.1007/s00216-019-01609-4). Thus, for bacterial studies, when the analytes such as bacterial pathogens adsorbed certain metal nanoparticles such as gold and silver, etc. on their surfaces, the signals could be enhanced by many orders of magnitude compared to normal RS (Nanostructured silver-gold bimetallic SERS substrates for selective identification of bacteria in human blood. doi: 10.1039/c3an01924a). In addition, SERS handled samples easily and provided the basis for non-destructive and ultra-sensitive detection of samples (Surface-enhanced Raman spectroscopy at single-molecule scale and its implications in biology. doi: 10.1098/rstb.2012.00262). Although we did not perform the normal RS for bacteria, previous studies had already done so and confirmed that normal RS could only give flat curves with no specific fingerprinting spectra and characteristic peaks (doi: 10.1007/s00216-019-01609-4, for specific figures, please see the Supplementary Figure S5 for the differences between RS and SERS). Taken together, SERS is a better method than Raman spectroscopy as a sensitive analytical tool. In the manuscript, we explained the reasons of why we need to use SERS for the study in the revised manuscript:

Thus, surface enhanced Raman spectroscopy (SERS) has been extensively developed to overcome the weak Raman scattering effect, which uses metallic nanoparticles (gold, silver and copper) to concentrate electromagnetic energy via surface plasmons (Pérez-Jiménez et al., 2020). For example, Witkowska et al. (2019) systematically compared the differences between RS and SERS in terms of bacterial detection, according to which SERS spectra had extremely better quality than the normal Raman spectra for both *Escherichia coli* and *Bacillus subtilis*.

- Why was normalization to the highest peak chosen? Which peak was the highest or does it differ for individual spectra?

Sorry for the misunderstanding. The description of the method in the manuscript is not accurate and mis-leading. Thus, we explained the detailed procedures here and also revised the part in the manuscript. In particular, for the matrix comprised of Raman shifts and intensities of all the CRKP spectra (n=280), each row is an independent Raman spectrum, while each column consists of the intensity values of all 280 spectra at a particular Raman shift. Raman intensities were vastly different between different Raman shifts, but generally similar within the same Raman shift. Through normalizing (compressing) all the intensity values in the same column in the range of 0 to 1, it could avoid too large intensity values and could also preserve too small intensity values in the averaged Raman spectrum. Otherwise, smaller Raman intensity values could become covered (near 0) in the averaged Raman spectrum. The details of the normalization method were described in the section 9.3 Feature Scaling via Standard Normalization of the book Machine Learning Refined: Foundations, Algorithms, and Applications (DOI: 10.1017/9781108690935).

In the manuscript, we revised this part as “In this study, we normalized SERS spectra by column (Raman intensity values at a particular Raman shift). That is, the highest intensity value (peak value) in each column was selected as the maximum constant, so that all the other measured spectral intensities in the same column were divided to the highest intensity value to realize the normalization of the spectral data (Watt et al., 2020).”

- Why the authors used unsupervised machine learning methods? They are commonly (as the authors declare as well) used for datasets without prior knowledge of all samples. Did authors try it that way, e.g. dividing the dataset to "known" and "unknown" (purposely "unidentified")?

At the beginning, what we wanted was to check whether we could discriminate CSKP and CRKP strains from each other by using unsupervised machine learning methods, in order to identify the intrinsic differences between the two groups. We did use unlabeled data for the unsupervised machine learning analysis. Later, when doing the supervised machine learning analysis, we divided the dataset into labeled (for training) and unlabeled data (for prediction). Cluster algorithms are normally divided into four types: partition clustering, hierarchical clustering, fuzzy clustering and density-based clustering. K-means, AGNES, and DBSCAN are the most common cluster algorithms among these types of algorithms (DOI: 10.1016/B978-044452701-1.00064-8). Thus, we used these three algorithms for our analysis. However, based on the reviewer's suggestion, we finally decided to remove all the unsupervised machine learning analysis from the manuscript since, as the reviewer's mentioned, they cannot predict the CSKP and CRKP strains without prior knowledge of all samples. Thus, there is no meaning to do the unsupervised machine learning analysis. In addition, by doing so, we also greatly reduced the lengths of the manuscript since the originally submitted version is too long.

- Suppl. Fig1: color coding is not 100% clear - authors use 3 levels of resistance, but four colors. Also, the color coding of rectangles dividing groups is a bit confusing - I suppose, most of the readers would expect green colored strains to be sensitive and red to be resistant, as in squares further in the chart...

Thanks for the reviewer's comments. We have changed the color for CSKP and CRKP strains in Supplementary Figure 1. As you can see now, CSKP was in green color while CRKP was in red color. As for color-coding squares, we rephrased our expression in the manuscript so as to avoid any confusions: Red (Carbapenem resistance) and blue (other antibiotic resistance) squares were used to represent different types of antibiotic resistance: true resistance (filled square), intermediate resistance (semi-transparent square), sensitivity (white square with border), and missing values (white square without border). (B) PCA analysis of *K. pneumoniae* strains based on antibiotics-resistant profiles. CRKP (red dots) and CSKP (green dots) were classified into two separate groups as indicated by the red and green circles, respectively.

(A)

(B)

Results:

- Fig 3: I would recommend the authors to add labels to the arrows so the differences are clearly markable.

We have already added the arrows to different characteristic peaks. However, due to the PDF generation process by the manuscript submission system, the arrows were lost. The original Fig. 3 was supplied here, which was showed below:

- Authors state: "As for CSKP strains, unique characteristic peaks were identified at 783 cm^{-1} (Thymine), 915660 cm^{-1} (C-C), 1328 cm^{-1} (Adenine ring) and 1636 cm^{-1} (Amide I), while unique characteristic peaks for CRKP strains were found at 890 cm^{-1} (Tryptophane), 1209 cm^{-1} (Phenylalanine/Tyrosine), 1244 cm^{-1} (Amide III), 1321 cm^{-1} (Guanine) and 1898 cm^{-1} (C=O)." - since the peak are marked as "unique characteristic peaks for CSKP/CRKP", how do authors explain their absence in CRKP/CSKP? Is there a possibility they are not correctly assigned to biomolecules? Or do the CRKP/CSKP strains have at least corresponding low intensity peaks? I believe the cells would need those components to survive.

Thanks for the reviewer's comments. As the reviewer pointed out, some of the biomacromolecules were essential for bacterial survival, but they were not shown by characteristic peaks. As the reviewer might see in Figure 3, the two spectra were very similar, which indicated the similar compositions between the two types of Kp strains. The reason for some of the characteristic peaks to be unique was due to the algorithm analysis which could be caused by the comparative low intensity peaks of certain compounds in the spectra, which should not be understood as absence of certain essential compounds in bacterial compositions.

- What is meant by accuracy and precision in the study? Do they apply to individual spectra or strains?

Accuracy is the overall judgment ability of the classifier, that is, the proportion of correct predictions. It does not consider whether the sample is positive or negative. Precision is the ratio of the number of samples correctly predicted by the classifier to the number of positive samples for all prediction positions, that is, how many of the samples predicted to be positive samples are truly positive samples. In general, Precision only focuses on the positive sample part, while accuracy considers all samples.

- How were the ellipses in PCA analysis depicted? Do they have any statistical relevance or were they drawn manually to include all points? (it also applies to suppl. figures)

There is no statistical relevance for the ellipses that were only used for demonstration purposes. The ellipses in both Supplementary Figure 1 and Supplementary Figure 3 were drawn manually to include as many points as possible.

- Were the incorrectly identified spectra acquired from one strain or more strains in each classification method?

Thanks for the reviewer's comment. The incorrectly identified spectra were acquired from one strain (strain 16-2) in Supplementary Figure 1.

Discussion

- Do authors believe that their method could distinguish between carbapenem-resistant and multidrug-resistant KP strains? Do authors think that the antibiotic profile as a whole reflects in Raman spectra or are the changes caused solely by carbapenem resistance?

We genuinely believe that the methods could discriminate CSKP from CRKP strains efficiently and accurately. However, this is a pilot study and the sample number is limited. In future, we need more CRKP and CSKP strains to train our model in order to make sure that our model has good performance with high robustness. In addition, since CRKP strains are generally extensively drug-resistant strains, it is not possible to isolate clinical Kp strains with only carbapenems resistance. Thus, the SERS spectra did reflect the whole antibiotic profiles rather than solely by carbapenem resistance. In future studies, we could try to construct Kp strains with only carbapenem resistance by carbapenem-resistant plasmid transformation.

- How would author explain the big difference between accuracies of unsupervised and supervised machine learning methods?

Thanks for the reviewer's comment. The different results between supervised and unsupervised machine learning methods could be explained as below. The cluster algorithms (unsupervised machine learning algorithm) are very sensitive to outliers and noise signals, and it is easy to reach local optimal solution. When the clustering distances are very different, the clustering qualities are comparatively poor. During processing the high-dimensional data like the Raman spectra, when the high-dimensional data is clustered to a low-dimensional space, a dimensional explosion will occur. In that case, when result is visualized, spectra from each strain will cluster into one group, but two clusters that are close to each other are unable to be separated. In contrast, the supervised machine learning algorithms can divide the data set into training set,

validation set and test set. We use machine learning and deep learning algorithms to train the model on the training set data, and then keep training the model on the validation set to adjust the model parameters in order to obtain the optimal model. After the optimal model is obtained, the model can recognize different types of data. In addition, since the unsupervised machine learning algorithms did not perform well in this study, and also because the unsupervised machine learning algorithms cannot provide any help in predicting CSKP and CRKP strains, we removed all the sections related to unsupervised machine learning algorithms so as to avoid any confusions and also to reduce the lengths of the manuscript.

- Discussion should include comparison of the results with other Raman-based studies of antimicrobial resistance in microbes.

The comparison of the results with other Raman-based studies of antimicrobial resistance in microbes were added to the discussion section:

1) In terms of the differentiation of antibiotic resistance and sensitivity in bacterial strains, a variety of studies have addressed this question. However, most of the studies used simple statistical models such as linear discriminant analysis (LDA) and principal component analysis (PCA) for data analysis. For example, Verma et al. used partial least squares-discriminant analysis (PLS-DA) to study Raman spectra of *Escherichia coli* strains treated with bacteriostatic and bactericidal antibiotics, which identified characteristic peaks that are altered by the antibiotic concentrations (Verma et al., 2020). Cheong et al. (2017) analyzed drop-coating deposition SERS spectra via PCA and SVM to identify quinolone-resistant *K. pneumoniae* strains.

2) Among these algorithms, CNN consistently performed best based on all the evaluation indicators (**Table 2**), ROC curves (**Figure 4**), and confusion matrix (**Figure 5**), achieving 99.78% accuracy during five-fold cross validation. Previously, Ho et al. (2019) used convolutional neural network (CNN) and support vector machine (SVM) to successfully identify methicillin-resistant *Staphylococcus aureus* (MRSA) and methicillin-sensitive *S. aureus* (MSSA) with an accuracy of 89±0.1%. Thus, the pilot study showed that CNN could be used for antibiotic resistance predictions in differential bacterial species with better performance. Moreover, By comparing with other machine learning methods used in this study, CNN could handle complex regression and classification problems without assumed mathematical equations between input and output, leading to high computational efficiency and strong fault tolerance (Wang et al., 2020).

- What do the authors plan for the future - do authors plan to perform more robust analysis of CRKP and CSKP strains? - What are the weak points of this study and how they could be improved?

Thanks for the reviewer's comments. We have added our future plan and discussed the weak points of this study in the discussion section: In this pilot study, we performed comparative analyses of supervised machine learning algorithms on discriminating CSKP and CRKP strains via SERS spectra. Although all the methods achieved high prediction accuracies, the models that we constructed were not robust and sufficient for real-world applications due to the limited number of *K. pneumoniae* strains used in this study. In addition, since both CSKP and CRKP strains were multi-drug resistant bacteria, other antibiotic resistances rather than carbapenem resistance may also be involved in the identification of CSKP and CRKP strains because of their contributions to the generation of SERS spectra. Thus, more SERS spectra from clinically isolated CSKP and CRKP strains should be used for training the machine learning models, which will greatly improve the quality and robustness of the models. In addition, antibiotic resistance profiles during *K. pneumoniae* isolation should be strictly controlled, and only

those strains with differences in carbapenem sensitivity and resistance were used for SERS spectral analysis while the profiles of other antibiotic resistance should be the same. In this way, machine learning models could predict CSKP and CRKP strains solely based on carbapenem resistance rather than other antibiotic resistances.

- How could the results contribute rapid diagnostic testing? At the moment, method seems to be time-consuming (necessary cultivation, further sample preparation for SERS, drying time), requiring trained personal and specific consumables. Could authors comment on this? What is the approximate time necessary for the identification?

We rephrased our expression, lowered our tone on fast diagnosis, and focused on the accurate prediction of CSKP and CRKP strains since the method did require bacterial culture, colony pick-up, and sample preparation (drying time). However, the method and procedure are really easy to acquire with no previous experience required. The acquisition time of the Raman signal for a signal spectrum only took 5-10 seconds. For the consumables, normally tips, EP tubes, and silicon wafer were needed, which were all cheap and easy to purchase. In addition, with further development of the technique, machine learning models based on large number of SERS spectra for CSKP and CRKP strains could be used to directly predict the presence and absence of CSKP and CRKP strains in the clinical samples, which will greatly facilitate the fast and accurate diagnosis of *K. pneumoniae* strains. In this discussion section of the revised manuscript, we wrote “It should be noted that, although acquisition of the signal for a single SERS spectrum took only seconds, the method used in this study still required bacterial culture and isolation, which made the overall procedure time-consuming. In future studies, we aim to use machine learning models to recognize CSKP and CRKP strains from clinical samples directly, which will greatly improve the efficiency of the rapid diagnostics of carbapenem-resistant *K. pneumoniae*.”

Additional comments:

-Abstract: "Conventional methods for 117 antibiotic resistance testing are sometimes time consuming while molecular methods..." - I wouldn't say sometimes since it is a bit confusing. Actually, they are time-consuming (in comparison to suggested method).

We have deleted the vague word “sometimes” in the abstract: Conventional methods for antibiotic resistance testing are time-consuming while molecular methods such as PCR-based testing might not accurately reflect phenotypic resistance.

-The article is extremely long, which could prevent readers from reading it whole - for example, the introduction is 4.5 pages long and contains a lot of information that are not necessarily relevant for the study or might fit better in the discussion. Also, several detailed descriptions from methods could be moved to the supplement. Plus, the texts are repeated in many sections...

We have shortened the introduction section to 2 pages. We also deleted the non-relevant part and moved some part of the introduction section to the discussion. In terms of the method section, we have removed most of the statistical descriptions due to their irrelevance to the topic of this study. We also went through the manuscript, deleting the repeated parts of the manuscript so as to improve the quality of the manuscript.

-Suppl. Fig. 1: Sentence "It was noteworthy that sample No. 16-2 was actually CRKP, which was mistakenly classified into the group of CSKP singly based on its antibiotics resistant phenotype." is not clear to me - since it had ATB resistant phenotype, why it was classified as CSKP based on it?

Thanks for the reviewer's question. As the reviewer can see from Supplementary Figure 1, strain 16-2 had similar carbapenems resistance profiles with other CRKP strains. However, 16-2, 20-18, and 18-88 showed similar antibiotics resistance profiles in the blue region (non-carbapenems resistance region), especially for Doxycycline, Cefotetan, and Tetracycline. After HCA analysis, the algorithm automatically assigned 16-2 into the CSKP group due to the profiles of non-carbapenems resistance, which outweighed the distributions of similar carbapenems resistance profiles.

-Suppl. Table 3: why are demographic data (gender, age) included? Are they relevant to the study? Were the samples anonymized?

These data were not relevant to the study. We removed them from the Supplementary Table 3. All the data used in this study were anonymized.

Reviewer #2 (Comments for the Author):

The article aims to demonstrate efficiency of different machine learning algorithms to discriminate among various strains of carbapenem-sensitive *Klebsiella pneumoniae* (CSKP) and carbapenem-resistant *Klebsiella pneumoniae* (CRKP) isolated from clinical samples. The data analysis of SERS spectra is well described and machine learning results are promising. In particular, according to the study, the authors identified unique profiles of characteristic peaks in average SERS spectra for CSKP and CRKP strains, respectively. In addition, they also concluded that convolutional neural network (CNN) could discriminate between CSKP and CRKP strains with high accuracy. There are also some other interesting findings reported during the study. Although the reviewer finds the study very interesting and promising, there are some issues that need to be addressed before the manuscript could be considered for publication in *Microbiology Spectrum*.

Many thanks for the reviewer's genuine comments, which greatly improved the quality of the manuscript. We have followed these comments and suggestions and revised the manuscript substantially. For details, please see our point-by-point responses below.

1. It would be better if the authors could add specific Raman shift for each black arrow in Figure 3 so it is much clearer to compare the differences of characteristic peaks between CSKP and CRKP strains. In addition, there should be figure legend showing which curve corresponded to which Kp strain, CSKP or CRKP.

Sorry for the confusion. We actually did all the labelling and arrow annotation. However, during the conversion from word format to PDF format by the submission system, some errors occurred that led to the confusing figure. We have fixed this issued and the new figure was present below for the reviewer's reference.

2. Could the authors explain why they choose to use the three clustering algorithms for CSKP and CRKP analysis?

Cluster algorithms are divided into four types: partition clustering, hierarchical clustering, fuzzy clustering and density-based clustering. K-means, AGNES, and DBSCAN are the most common cluster algorithms among these types of algorithms (DOI: 10.1016/B978-044452701-1.00064-8). Thus, we used these three algorithms for our analysis. However, we have decided to remove the unsupervised machine learning part in the manuscript since it can only tell the clustering effects without facilitating the recognition and discrimination of antibiotic resistances in CSKP and CRKP strains.

3. Could the authors provide explanations for the capacity of CNN algorithm in terms of its better stability and accuracy for low SNR SERS spectra?

Thanks for the reviewer's question. The convolutional neural network has a multi-layer structure such as a convolutional layer and a pooling layer, which makes the convolutional neural network have translational invariance. In addition, when the data was affected by noise, the largest pooling layer used in this study showed non-deformation feature, which made sure

that the feature map did not change much during the analysis process. Thus, due to such features, the CNN algorithm had better robustness and noise resistance when compared with other methods.

4. Please make sure that all the references in the reference list follow the same format. For example, please use full name or abbreviation for all the journal names.

We have revised the reference list as required by the reviewer and all the journal names have been double-checked. Full journal names were used for all the references. For details, please refer to the re-submitted version of the manuscript. (Please see the reference list attached at the bottom of this letter).

5. Have the authors tried other types of Raman spectrometer for the experiment, such as handheld Raman spectrometer, which will be more convenient in detecting bacterial pathogens as a POCT device and have more applications in real-world situations?

Thanks for the reviewer's comment. Although we did not use the portable Raman spectrometer to test the samples of CSKP and CRKP in this study, we recently did a pilot study on *Mycobacterium tuberculosis* by using the handheld Raman spectrometer (Cora100, Anton Paar, Austria) and successfully separated *M. tuberculosis* strains isolated from different patients. The averaged Raman spectra for these *M. tuberculosis* strains were generated and were shown below in **Figure 1**.

Figure 1 Averaged SERS spectra for 8 *Mycobacterium tuberculosis* strains isolated from different patients (unpublished data)

As you can see from the SERS spectra, the standard error bands were really wide, which indicated that the repeatability of the spectra was not as good as the desktop Raman

spectrophotometer. We also did cluster analysis for these spectra via OPLS-DA method, which showed that these spectra could be well separated into different clusters. Thus, although handheld Raman spectrophotometer showed comparatively low reproducibility, the results were still promising since we could discriminate *M. tuberculosis* strains from different isolates.

Figure 2 Cluster analysis of SERS spectra that were generated via handheld Raman spectrophotometer through OPLS-DA method (unpublished data)

Thus, based on this pilot study, we are assured that we will perform more studies on antimicrobial analysis and bacterial identifications by using handheld Raman spectrometer.

6. The methods and materials section is a bit long and the authors may want to shorten this section, especially for the routine procedures of Raman spectral preprocessing.

Thanks for the reviewer's suggestions. We have addressed this issue and reduced the manuscript substantially. For details, please see the re-submitted revised manuscript.

December 27, 2021

Dr. Liang Wang
Xuzhou Medical University
School of Medical Informatics
No.209, Tongshan Road
Yunlong District
Xuzhou, Jiangsu 221000
China

Re: Spectrum02409-21R1 (Discrimination between carbapenem-resistant and carbapenem-sensitive *Klebsiella pneumoniae* strains through computational analysis of surface enhanced Raman spectra: a pilot study)

Dear Dr. Liang Wang:

Thank you for significantly modifying your original manuscript based upon the reviewers' recommendations. While the revised manuscript is significantly improved, one of the reviewers has requested additional edits with which I agree. Once these issues are addressed I believe the manuscript can move forward for publication.

Link Not Available

Sincerely,

Karen Carroll

Journals Department
Reviewer comments:

Reviewer #1 (Comments for the Author):

I appreciate that authors reworked the file and made it shorter and more clear. However, from my point of view, it certain parts it is still too wordy. Further shortening might enhance readability and fluency of the paper.

Examples:

- Introduction: the first paragraph could be concentrated into 1-2 sentences... Contrariwise, a few sentences regarding SERS (the basic principle) would be appropriate since not all readers must know the technique...
- Discussion: now, the discussion is quite long - it includes comparisons of the results from different points of views: clinical diagnostics, SERS, machine learning methods, baseline correction. From my point of view, it is overwhelming. Together with wordy sentences and jumps between the topics, the reader might get lost.

Authors should consider cutting the clutter or narrowing the spectrum of discussed topics to a clear goal.

Methods:

- To avoid any potential (regulatory or confidentiality) concerns, the authors should state that they worked with anonymized samples and/or just bacterial isolates, not a specimen itself - it is not clear from the description in the methods section
- Since the authors use rpm not rcf, type of the centrifuge (rotor) should be specified
- Numbers of spectral acquisitions using SERS: the authors state (in the comments to the review) that not all acquired spectra were at sufficient quality, therefore the numbers of acquisitions are not equal for all strains, could authors comment on the process of spectral acquisition in more details? Is the problem visible at the moment of spectral acquisition or later during the processing of spectra? How were they defined? Were those problematic spectra included in the analysis? I think that this might be necessary for future clinical use, since, as the authors say, there might be problems with repeatability when using SERS...

Results:

- suppl. Fig. 3: However, it was noteworthy that sample No. 26-272 was CSKP but mistakenly identified as CSKP. - I believe there is a typo in the sentence otherwise it does not make a sense... How was the PCA performed? Using the whole spectrum or just some peaks? I believe that in this case, it would be necessary to depict ellipses with some statistical significance (e.g. CI 90%) otherwise, it is confusing and misleading.
- I would highly recommend the authors to remove peak assignments from brackets on the page 12 (specific peaks for CSKP and CRKP) or to comment on the fact that they are not present in the spectrum of the CSKP or CRKP - as the authors wrote in the comments to the review "The reason for some of the characteristic peaks to be unique was due to the algorithm analysis which could be caused by the comparative low intensity peaks of certain compounds in the spectra, which should not be understood as absence of certain essential compounds in bacterial compositions." I agree with that plus the assignments are suggested, therefore the current presentation in the text might lead to confusion.

Reviewer #2 (Comments for the Author):

The authors have addressed my comments and I agree acceptance.

Staff Comments:

Preparing Revision Guidelines

Please return the manuscript within 60 days; if you cannot complete the modification within this time period, please contact me. If you do not wish to modify the manuscript and prefer to submit it to another journal, please notify me of your decision immediately so that the manuscript may be formally withdrawn from consideration by Microbiology Spectrum.

I appreciate that authors reworked the file and made it shorter and more clear. However, from my point of view, it certain parts it is still too wordy. Further shortening might enhance readability and fluency of the paper.

Examples:

- Introduction: the first paragraph could be concentrated into 1-2 sentences... Contrariwise, a few sentences regarding SERS (the basic principle) would be appropriate since not all readers must know the technique...

- Discussion: now, the discussion is quite long – it includes comparisons of the results from different points of views: clinical diagnostics, SERS, machine learning methods, baseline correction. From my point of view, it is overwhelming. Together with wordy sentences and jumps between the topics, the reader might get lost.

Authors should consider cutting the clutter or narrowing the spectrum of discussed topics to a clear goal.

Methods:

- To avoid any potential (regulatory or confidentiality) concerns, the authors should state that they worked with anonymized samples and/or just bacterial isolates, not a specimen itself – it is not clear from the description in the methods section
- Since the authors use rpm not rcf, type of the centrifuge (rotor) should be specified
- Numbers of spectral acquisitions using SERS: the authors state (in the comments to the review) that not all acquired spectra were at sufficient quality, therefore the numbers of acquisitions are not equal for all strains, could authors comment on the process of spectral acquisition in more details? Is the problem visible at the moment of spectral acquisition or later during the processing of spectra? How were they defined? Were those problematic spectra included in the analysis? I think that this might be necessary for future clinical use, since, as the authors say, there might be problems with repeatability when using SERS...

Results:

- suppl. Fig. 3: However, it was noteworthy that sample No. 26-272 was CSKP but mistakenly identified as CSKP. – I believe there is a typo in the sentencem otherwise it does not make a sense... How was the PCA performed? Using the whole spectrum or just some peaks? I believe that in this case, it would be necessary to depict ellipses with some statistical significance (e.g. CI 90%) otherwise, it is confusing and misleading.
- I would highly recommend the authors to remove peak assignments from brackets on the page 12 (specific peaks for CSKP and CRKP) or to comment on the fact that they are not present in the spectrum of the CSKP or CRKP – as the authors wrote in the comments to the review "The reason for some of the characteristic peaks to be unique was due to the algorithm analysis which could be caused by the comparative low intensity peaks of certain compounds in the spectra, which should not be understood as absence of certain essential compounds in bacterial compositions." I agree with that plus the assignments are suggested, therefore the current presentation in the text might lead to confusion.

Point-by-Point Responses to Editorial and Reviewers' Comments

Manuscript No. Spectrum02409-21R1

Dec. 30th, 2021

Reviewers' comments

Reviewer #1 (Comments for the Author):

I appreciate that authors reworked the file and made it shorter and more clear. However, from my point of view, in certain parts it is still too wordy. Further shortening might enhance readability and fluency of the paper.

Many thanks for the reviewer's constructive comment. We agree with the reviewer and have further shortened the manuscript in order to enhance the readability and fluency of the manuscript.

Examples:

Introduction: the first paragraph could be concentrated into 1-2 sentences... Contrariwise, a few sentences regarding SERS (the basic principle) would be appropriate since not all readers must know the technique...

The first paragraph has been re-written and then shortened to 3 sentences as shown below: "Many microbial organisms are pathogenic to human beings and are able to cause infectious diseases (1). In addition, drug-resistant bacterial pathogens have been emerging due to the overuse and misuse of antibiotics (2), which leads to difficulty in bacterial control and imposes further threats upon global public health. Thus, fast and accurate detection of antibiotics-resistant bacteria is necessary for clinical treatment of bacterial infection and prevention of bacterial transmission (3)."

In the last paragraph of the Introduction section, we added a few sentences regarding basic principles of SERS:

"Surface enhanced Raman spectroscopy (SERS) is a non-destructive chemical analysis technique that could improve the weak signals of regular Raman spectroscopy through interactions between sample molecules and surface plasmons of nanoscale-structured metal particles (15). In particular, signal-enhancing metal nanostructures such as silver (Ag), copper (Cu) and gold (Au) can generate a plasmon resonance electromagnetic enhancement of the stimulating light, which could greatly increase the signal level of Raman spectroscopy up to several orders of magnitude (16). However, due to the complexity of Raman spectra, traditional linear analysis is not sufficient for the data processing procedures while machine learning algorithms are capable of extracting important features from the sophisticated SERS spectral datasets (15, 16). Thus, SERS provides a great potential for fast and sensitive microbial detection and identification with the assistance of appropriate machine learning (ML) algorithms (17)."

Discussion: now, the discussion is quite long - it includes comparisons of the results from different points of views: clinical diagnostics, SERS, machine learning methods,

baseline correction. From my point of view, it is overwhelming. Together with wordy sentences and jumps between the topics, the reader might get lost. Authors should consider cutting the clutter or narrowing the spectrum of discussed topics to a clear goal. Thanks for the reviewer's comments, which we totally agree with. Thus, we decide to focus our discussion on the computational discrimination of CSKP and CRKP strains via SERS spectra since the main theme of this study as reflected in the manuscript title was "Discrimination between carbapenem-resistant and carbapenem-sensitive *Klebsiella pneumoniae* strains through computational analysis of surface enhanced Raman spectra". Although machine learning algorithms and comparison of other statistical analyses were also important parts in this study, we decided to reduce the corresponding contents after careful consideration of the readership of Microbiology Spectrum. Spectra pre-processing procedures were also removed to make our discussion achieving a much clearer goal. For details of the revision, please refer to our re-submitted marked-up manuscript (The total number of words for the manuscript including reference is 7701, which should meet the editorial and reviewers' requirements for shortening the manuscript).

Methods:

1. -To avoid any potential (regulatory or confidentiality) concerns, the authors should state that they worked with anonymized samples and/or just bacterial isolates, not a specimen itself - it is not clear from the description in the methods section.

We added a sentence in Section 2.1 "It is noteworthy that all the clinical samples were previously de-identified and only bacterial isolates were analyzed in this study."

2. -Since the authors use rpm not rcf, type of the centrifuge (rotor) should be specified For stirring, a magnetic stirrer (Model No. ZNCL-BS230, Shi-Ji-Hua-Ke Pty. Ltd., Beijing, China) was used at the speed of 650 r/min. As for the centrifuge step, Eppendorf centrifuge 5430 R was used, which was specified in Section 2.2 as requested (Centrifuge 5430 R, Eppendorf, United States).

3. -Numbers of spectral acquisitions using SERS: the authors state (in the comments to the review) that not all acquired spectra were at sufficient quality, therefore the numbers of acquisitions are not equal for all strains, could authors comment on the process of spectral acquisition in more details? Is the problem visible at the moment of spectral acquisition or later during the processing of spectra? How were they defined? Were those problematic spectra included in the analysis? I think that this might be necessary for future clinical use, since, as the authors say, there might be problems with repeatability when using SERS...

Thanks for the reviewer's comment. During the data collection stage of the raw SERS spectra, it was actually difficult to justify the quality of the technical repeats of spectra. In addition, for the technical repeats of each sample, laser beam was moved around the dried spot of bacterial sample for each single measurement, which would make sure that all the SERS spectra gave an overall description of the sample, even when the

quality of some spectra was not good (large error band). The procedure was described in the manuscript “[After each *K. pneumoniae* strain was cultured on the agar plate overnight, a single colony was picked up and inoculated into 15 \$\mu\$ L phosphate buffer saline \(PBS\) with vigorously vortexing. 15 \$\mu\$ L of negatively-charge AgNPs was then mixed with the PBS solution, which was then dropped onto a silicon wafer for air dry. The dried spot was then measured via commercial i-Raman[®] PLUS Raman spectrometer BWS456-785H \(B&W Tek, USA\)](#)”. After multiple times of measurement, we exported the raw data in CSV format from the Raman spectroscopy via software, then combined these technical repeats together, and finally used average SRES spectrum and error band to check the overall quality of the experimental reproducibility, quantitatively. There are also some other statistical methods that we can use to check the quality of the SERS spectra from the same clinical sample, such as different types of clustering analysis. However, the intrinsic low reproducibility did hinder the clinical application of Raman spectroscopy, which is currently under extensive investigations and drives many researchers to work on the issue. In practice, there are many factors that could influence the reproducibility of the experiment (<https://doi.org/10.1007/s12204-014-1566-7>) such as the distribution of metal particles on the surface of samples and fluctuation of measurement conditions (attenuation of laser power, temperature-variation induced impact on the photon, detection efficiency of charge-coupled device, the alignment-dependent spot size of laser focus, excitation efficiency, and transmission efficiency of the spectrograph, etc.). All these factors could lead to the variation of SERS spectra, which is why we need to measure the sample for multiple times the same sample. All the SERS spectra were finally used for the CSKP and CRKP analysis. Although the number of SERS spectra varies within the group, the number of SERS spectra for CSKP and CRKP was the same, which made further machine learning analysis feasible and rational. In addition, in order to make sure that our computational analysis was reliable, we preprocessed the spectral signal through a series of methods, and then used 5-fold cross-validation to verify whether these data would affect our prediction results. The cross-validation score for CNN was 99.78%, which confirmed that these data had little impact on the performance of machine learning models. In our future studies, in order to make confusions to the readership, we will definitely use equal number of spectra within the group for each bacterial isolate.

Results:

Suppl. Fig. 3: However, it was noteworthy that sample No. 26-272 was CSKP but mistakenly identified as CSKP. - I believe there is a typo in the sentencem otherwise it does not make a sense.

Sorry for the typo. We have already revised the mistake in the supplementary material as following: “However, it was noteworthy that sample No.26-272 was **CRKP** but mistakenly identified as CSKP”.

How was the PCA performed? Using the whole spectrum or just some peaks? I believe that in this case, it would be necessary to depict ellipses with some statistical significance (e.g. CI 90%) otherwise, it is confusing and misleading.

In the manuscript, we wrote “In addition, PCA analysis was performed based on the distribution of characteristic peaks of each bacterial strain in order to separate all the *K. pneumoniae* strains to different groups (Supplementary Figure 3).” It should be noted that the two ellipses in Supplementary Figure 3 were manually added to distinguish the two clusters of data.

In specificity, we used scatter plot to demonstrate the two groups of SERS spectra via their distributions of characteristic peaks. The purpose was to see whether the two groups of *K. pneumoniae* strains could be separated and whether the differences within the same group were small. All the data used for the clustering visualization were generated through dimensionality reduction by PCA algorithm. After PCA analysis, the data were divided into 2 categories and visualized. The data clustering was largely matched with their antibiotics-resistant phenotype. In addition, through PCA analysis, the data dimensions were reduced, decreasing the computational workload of machine learning analysis and increasing the prediction capacity of machine learning models. Similar procedures were previously used by many other studies like doi: 10.1128/AEM.00924-20 for Raman spectral clustering analysis.

In order to answer the reviewer's question in terms of the eclipse with statistical significance, we performed the PCA analysis and visualized the data with 95% CI, which showed that the two groups could not be well separated. Thus, we kept the Supplementary Figure 3 in the supplementary material to show that the two groups could be well separated.

-I would highly recommend the authors to remove peak assignments from brackets on the page 12 (specific peaks for CSKP and CRKP) or to comment on the fact that they are not present in the spectrum of the CSKP or CRKP - as the authors wrote in the comments to the review "The reason for some of the characteristic peaks to be unique was due to the algorithm analysis which could be caused by the comparative low intensity peaks of certain compounds in the spectra, which should not be understood as absence of certain essential compounds in bacterial compositions." I agree with that plus the assignments are suggested, therefore the current presentation in the text might lead to confusion.

We agree with the reviewer and the part of specific peaks for CSKP and CRKP has been removed "~~As for CSKP strains, unique characteristic peaks were identified at 783 cm^{-1} (Thymine), 915 cm^{-1} (C-C), 1328 cm^{-1} (Adenine ring) and 1636 cm^{-1} (Amide I), while unique characteristic peaks for CRKP strains were found at 890 cm^{-1} (Tryptophane); 1209 cm^{-1} (Phenylalanine/Tyrosine), 1244 cm^{-1} (Amide III), 1321 cm^{-1} (Guanine) and 1898 cm^{-1} (C=O)."~~

Reviewer #2 (Comments for the Author):

The authors have addressed my comments and I agree acceptance.

Many thanks for the reviewer's efforts for improving the quality of the manuscript.

January 5, 2022

Dr. Liang Wang
Xuzhou Medical University
School of Medical Informatics
No.209, Tongshan Road
Yunlong District
Xuzhou, Jiangsu 221000
China

Re: Spectrum02409-21R2 (Discrimination between carbapenem-resistant and carbapenem-sensitive *Klebsiella pneumoniae* strains through computational analysis of surface enhanced Raman spectra: a pilot study)

Dear Dr. Liang Wang:

Your manuscript has been accepted, and I am forwarding it to the ASM Journals Department for publication. You will be notified when your proofs are ready to be viewed.

Sincerely,

Karen Carroll
Editor, Microbiology Spectrum
